# Goal-Conditioned Predictive Coding for Offline Reinforcement Learning

**Zilai Zeng**
Brown University

**Ce Zhang**
Brown University

**Shijie Wang**
Brown University

**Chen Sun**
Brown University

## Abstract

Recent work has demonstrated the effectiveness of formulating decision making as supervised learning on offline-collected trajectories. Powerful sequence models, such as GPT or BERT, are often employed to encode the trajectories. However, the benefits of performing sequence modeling on trajectory data remain unclear. In this work, we investigate whether sequence modeling has the ability to condense trajectories into useful representations that enhance policy learning. We adopt a two-stage framework that first leverages sequence models to encode trajectory-level representations, and then learns a goal-conditioned policy employing the encoded representations as its input. This formulation allows us to consider many existing supervised offline RL methods as specific instances of our framework. Within this framework, we introduce Goal-Conditioned Predictive Coding (GCPC), a sequence modeling objective that yields powerful trajectory representations and leads to performant policies. Through extensive empirical evaluations on AntMaze, FrankaKitchen and Locomotion environments, we observe that sequence modeling can have a significant impact on challenging decision making tasks. Furthermore, we demonstrate that GCPC learns a goal-conditioned latent representation encoding the future trajectory, which enables competitive performance on all three benchmarks. Our code is available at https://brown-palm.github.io/GCPC/.

## 1 Introduction

Goal-conditioned imitation learning [13, 15, 20] has recently emerged as a promising approach to solve offline reinforcement learning problems. Instead of relying on value-based methods, they directly learn a policy that maps states and goals (e.g. expected returns, or target states) to the actions. This is achieved by supervised learning on offline collected trajectories (i.e. sequences of state, action, and reward triplets). This learning paradigm enables the resulting framework to avoid bootstrapping for reward propagation, a member of the "Deadly Triad" [45] of RL which is known to lead to unstable optimization when combining with function approximation and off-policy learning, and also allows the model to leverage large amounts of collected trajectories and demonstrations. The emergence of RL as supervised learning on trajectory data coincides with the recent success of the Transformer architecture [46, 20] and its applications to sequence modeling, such as GPT [8] for natural language and VPT [3] for videos. Indeed, several recent works have demonstrated the effectiveness of sequence modeling for offline RL [10, 27, 34, 44, 49, 9, 42]. Most of these approaches apply the Transformer architecture to jointly learn the trajectory representation and the policy. When the sequence models are unrolled into the future, they can also serve as "world models" [21, 22] for many RL applications. By leveraging advanced sequence modeling objectives from language and visual domains [11, 4, 24], such methods have shown competitive performance in various challenging tasks [16, 5].

Despite the enthusiasm and progress, the necessity of sequence modeling for offline reinforcement learning has been questioned by recent work [15]: With a simple but properly tuned neural architecture (e.g. a multilayer perceptron), the trained agents can achieve competitive performance on several

37th Conference on Neural Information Processing Systems (NeurIPS 2023).

challenging tasks while taking only the current state and the overall goal as model inputs. In some tasks, these agents even significantly outperform their counterparts based on sequence modeling. These observations naturally motivate the question: Is explicit sequence modeling of trajectory data necessary for offline RL? And if so, how should it be performed and utilized?

To properly study the impact of sequence modeling for decision making, we propose to decouple trajectory representation learning and policy learning. We adopt a two-stage framework, where sequence modeling can be applied to learn the trajectory representation, the policy, or both. The connection between the two stages can be established by leveraging encoded trajectory representations for policy learning, or by transferring the model architecture and weights (i.e. the policy can be initialized with the same pre-trained model used for trajectory representation learning). These stages can be trained separately, or jointly and end-to-end. This design not only facilitates analyzing the impacts of sequence modeling, but is also general such that prior methods can be considered as specific instances, including those that perform joint policy learning and trajectory representation learning [10, 34, 49], and those that learn a policy directly without sequence modeling [15].

Concretely, we aim at investigating the following questions: (1) Are offline trajectories helpful due to sequence modeling, or simply by providing more data for supervised policy learning? (2) What would be the most effective trajectory representation learning objectives to support policy learning? Should the sequence models learn to encode history experiences [10], future dynamics [27], or both [19]? (3) As the same sequence modeling framework may be employed for both trajectory representation learning and policy learning [34, 9, 49], should they share the same training objectives or not?

We design and conduct experiments on the AntMaze, FrankaKitchen, and Locomotion environments to answer these questions. We observe that sequence modeling, if properly designed, can effectively aid decision-making when its resulting trajectory representation is used as an input for policy learning. We also find that there is a discrepancy between the optimal self-supervised objective for trajectory representation learning, and that for policy learning. These observations motivate us to propose a specific design of the two-stage framework: It compresses trajectory information into compact bottlenecks via sequence modeling pre-training. The condensed representation is then used for policy learning with a simple MLP-based policy network. We observe that goal-conditioned predictive coding (**GCPC**) is the most effective trajectory representation learning objective. It enables competitive performance across all benchmarks, particularly for long-horizon tasks. We attribute the strong empirical performance of GCPC to the acquisition of goal-conditioned latent representations about the future, which provide crucial guidance for decision making.

To summarize, our main contributions include:

- We propose to decouple sequence modeling for decision making into a two-stage framework, namely trajectory representation learning and policy learning. It provides a unified view for several recent reinforcement learning via supervised learning methods.

- We conduct a principled empirical exploration with our framework to understand if and when sequence modeling of offline trajectory data benefits policy learning.

- We discover that goal-conditioned predictive coding (GCPC) serves as the most effective sequence modeling objective to support policy learning. Our overall framework achieves competitive performance on AntMaze, FrankaKitchen and Gym Locomotion benchmarks.

## 2 Related Work

**What is essential for Offline RL?** Offline reinforcement learning aims to obtain effective policies by leveraging previously collected datasets. Prior work usually adopts dynamic programming [31, 29, 18] or supervised behavioral cloning (BC) methods [43, 3]. Recent approaches [10, 27] demonstrate the effectiveness of solving decision making tasks with sequence modeling, whereas RvS [15] establishes a strong MLP baseline for conditional behavioral cloning. To further understand what are the essential components for policy learning, researchers have investigated the assumptions required to guarantee the optimality of return-conditioned supervised learning [7], and examined offline RL agents from three fundamental aspects: representations, value functions, and policies [17]. Another concurrent work [6] compares the preferred conditions (e.g. data, task, and environments) to perform Q-learning and imitation learning. In our work, we seek to understand how sequence modeling may benefit offline RL and study the impacts of different sequence modeling objectives.

**Transformer for sequential decision-making.** Sequential decision making has been the subject of extensive research over the years. The tremendous success of the Transformer model for natural language processing [11, 40] and computer vision [24, 35] has inspired numerous works that seek to apply such architectures for decision making, and similarly motivates our work. Prior work has shown how to model sequential decision making as autoregressive sequence generation problem to produce a desired trajectory [10], while others have explored the applications of Transformer for model-based RL [27, 47] and multi-task learning [41, 9]. Our work aims to utilize Transformer-based models to learn good trajectory representations that can benefit policy learning.

**Masked Autoencoding.** Recent work in NLP and CV has demonstrated masked autoencoding (MAE) as an effective task for self-supervised representation learning [11, 8, 4, 24]. Inspired by this, Uni[MASK] [9], MaskDP [34], and MTM [49] propose to train unified masked autoencoders on trajectory data by randomly masking the states and actions to be reconstructed. These models can be directly used to solve a range of decision making tasks (e.g. return-conditioned BC, forward dynamics, inverse dynamics, etc.) by varying the masking patterns at inference time, without relying on task-specific fine-tuning. In contrast, we decouple trajectory representation learning and policy learning into two stages. Unlike approaches which perform value-based offline RL (e.g. TD3, Actor-Critic) after pretraining [34, 49], our policy learning stage adopts imitation learning setup where the policy can be learned without maximizing cumulative reward.

**Self-supervised Learning for RL.** Self-supervised learning has emerged as a powerful approach for learning useful representations in various domains, including reinforcement learning (RL). In RL, self-supervised learning techniques aim to leverage unlabeled or partially observed data to pre-train agents or learn representations that facilitate downstream RL tasks. Prior work mainly focuses on using self-supervised learning to get state representations [51, 38, 36, 50, 10, 34, 9, 49, 32] or world models [22, 23, 44, 27, 12, 37]. In [51], the authors evaluate a broad set of state representation learning objectives on offline datasets and demonstrate the effectiveness of contrastive self-prediction. In this work, we investigate representation learning objectives in trajectory-space with the sequence modeling tool, which enables us to explore the impact of different modalities (e.g. states, actions, goals, etc.) on trajectory representation learning.

## 3 Method

We revisit the role of sequence modeling in offline reinforcement learning, from the perspectives of trajectory representation learning and policy learning. Section 3.1 describes the background knowledge on extracting policies from offline trajectories using supervised learning. Section 3.2 introduces a two-stage framework that decouples trajectory representation learning and policy learning, which serves as the basis for our investigation. In Section 3.3, we propose a specific instantiation of the two-stage framework, in which we leverage a self-supervised learning objective – Goal-Conditioned Predictive Coding (**GCPC**), to acquire performant trajectory representations for policy learning.

### 3.1 Offline Reinforcement Learning via Supervised Learning

We follow prior work that leverages sequence modeling for offline reinforcement learning, and adopt the Reinforcement learning via Supervised learning (RvS) (e.g. [15]) setting. RvS aims to solve the offline RL problem as conditional, filtered, or weighted imitation learning. It assumes that a dataset has been collected offline, but the policies used to collect the dataset may be unknown, such as with human experts or a random policy. The dataset contains a set of trajectories: $\mathcal{D} = \{\tau^i\}_{i=1}^N$. Each trajectory $\tau^i$ in the dataset is represented as $\{(s_t, a_t)\}_{t=1}^H$, where $H$ is the length of the trajectory, and $s_t$, $a_t$ refer to the state, action at timestep $t$, respectively. A trajectory may optionally contain the reward $r_t$ received at timestep $t$.

As the trajectories are collected with unknown policies, they may not be optimal or have expert-level performance. Prior work [10, 33, 30] has shown that properly utilizing offline trajectories containing suboptimal data might produce better policies. Intuitively, the suboptimal trajectories may still contain sub-trajectories that demonstrate useful "skills", which can be composed to solve new tasks.

Given the above trajectories, we denote a goal-conditioned policy $\pi_\theta$ as a function parameterized by $\theta$ that maps an observed trajectory $\tau_{\text{obs}}^{(t)} = \{s_{\leq t}, a_{<t}\}$ and a goal $g^{(t)}$ to an action $a_t$. The goal $g^{(t)}$ is computed from $\tau_{t:H}$, which can be represented as either a target state configuration sampled

from $s_{t:H}$ or an expected cumulative reward/return-to-go computed from $r_{t:H}$ (see Appendix A.6 for details). For simplicity of notation, we write $\tau_{\text{obs}}^{(t)}$ as $\tau_{\text{obs}}$ and $g^{(t)}$ as $g$. We consider a policy should be able to take any form of state or trajectory information as input and predict the next action: (1) when only the current observed state $s_t$ and the goal $g$ are used, the policy $\hat{a}_t = \pi_\theta(s_t, g)$ ignores the history observations; (2) when $\pi_\theta$ is a sequence model, it can employ the whole observed trajectory to predict the next action $\hat{a}_t = \pi_\theta(\tau_{\text{obs}}, g)$. To optimize the policy, a commonly used objective is to find the parameters that fit the mapping of observations to actions using maximum likelihood estimation:

$$\theta^* = \underset{\theta}{\arg\max}\, \mathbb{E}_{\tau \sim \mathcal{D}} \left[ \prod_{t=1}^{|\tau|} \pi_\theta(a_t | \tau_{\text{obs}}, g) \right] \tag{1}$$

## 3.2 Decoupled Trajectory Representation and Policy Learning

Sequence modeling can be used for decision making from two perspectives, namely trajectory representation learning and policy learning. The former aims to acquire useful representations from raw trajectory inputs, often in the form of a condensed latent representation, or the pretrained network weights themselves. The latter aims to map the observations and the goal into actions that accomplish the task. To explicitly express the trajectory representation function, we rewrite the goal-conditioned policy function as follows:

$$\hat{a}_t = \pi_\theta(f(\tau_{\text{obs}}), g) \tag{2}$$

where $f(\cdot)$ is an arbitrary function, such as a neural network, that computes representation from $\tau_{\text{obs}}$. $f(\cdot)$ can also be an identity mapping such that $\pi_\theta(\cdot)$ directly operates on $\tau_{\text{obs}}$.

Motivated by the recent success of sequence modeling in NLP [11, 40] and CV [24, 14], both the trajectory learning function and the policy learning function can be implemented with Transformer neural networks as $f_\phi(\cdot)$ and $\pi_\theta(\cdot)$. We hypothesize that it is beneficial for $f_\phi$ to condense the trajectories into a compact representation using sequence modeling techniques. We also hypothesize that it is desirable to decouple the trajectory representation learning from policy learning. The decoupling not only offers flexibility on the choice of representation learning objectives, but also allows us to study the impact of sequence modeling for trajectory representation learning and policy learning independently. We thus adopt a two-stage framework with a TrajNet $f_\phi(\cdot)$ and a PolicyNet $\pi_\theta(\cdot)$. TrajNet aims to learn trajectory representations with self-supervised sequence modeling objectives, such as masked autoencoding [24] or next token prediction [40]. PolicyNet aims to obtain a performant policy with the supervised learning objective in Equation 1.

We now describe an overall flow of the two networks, which we will show is general to represent recently proposed methods. In the first stage, we approach trajectory representation learning as masked autoencoding. TrajNet receives a trajectory $\tau$ and an optional goal $g$, and is trained to reconstruct $\tau$ from a masked view of the same. Optionally, TrajNet also generates a condensed trajectory representation $B$, which can be utilized by PolicyNet for subsequent policy learning:

$$\hat{\tau}, B = f_\phi(\texttt{Masked}(\tau), g) \tag{3}$$

During the second stage, TrajNet is applied on the unmasked observed trajectory $- f_\phi(\tau_{\text{obs}}, g)$, to obtain $B_{\text{obs}}$. PolicyNet then predicts the action $a$ given $\tau_{\text{obs}}$ (or the current observed state $s_t$), the goal $g$, and trajectory representation $B_{\text{obs}}$:

$$a = \pi_\theta(\tau_{\text{obs}}, B_{\text{obs}}, g) \tag{4}$$

Our framework provides a unified view to compare different design choices (e.g. input information, architecture, training objectives, etc.) for representation learning and policy learning, respectively. Many existing methods can be seen as special cases of our framework. For example, prior works [34, 49] instantiate TrajNet with a bi-directional Transformer pre-trained via masked prediction, and re-use it as the PolicyNet in a zero-shot manner. To implement DT [10], $f(\cdot)$ is set as an identity mapping function of the input trajectory and $\pi(\cdot)$ is trained to autoregressively generate actions. These methods learn trajectory representations and the policy jointly, with training objectives inspired by MAE [24] and GPT [40]. At last, our framework recovers RvS-G/R [15] by having the output of $f(\cdot)$ in Eq. 2 as the last observed state $s_t$ from $\tau_{\text{obs}}$.

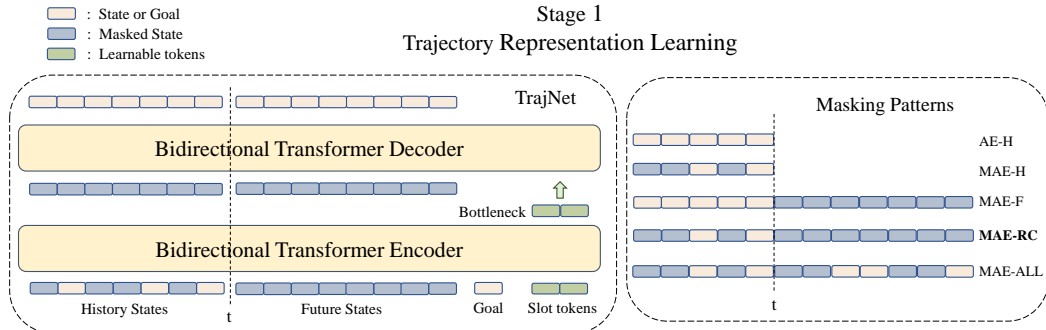

Figure 1: **Stage 1 – Trajectory Representation Learning.** For notation consistency between the two stages, we separate the input into observed (history) states and future states. **Left: TrajNet.** We input randomly masked history state, goal and slot tokens to the transformer encoder. The decoder takes in encoded slot tokens (the bottleneck) and a sequence of masked tokens, and reconstruct the whole trajectory. With this training objective, we encourage the bottleneck to perform predictive coding, which is conditioned on the goal and history states. **Right: All Masking Patterns.** TrajNet can be trained with different masking patterns with their corresponding reconstruction objectives. The illustration of TrajNet on the left uses "MAE-RC", which is adopted by GCPC.

## 3.3  Goal-Conditioned Predictive Coding

Finally, we introduce a specific design of the two-stage framework, Goal-Conditioned Predictive Coding (GCPC), which is inspired by our hypotheses introduced in Section 3.2. To facilitate the transfer of trajectory representations between the two stages, we compress the trajectory into latent representations using sequence modeling, which we refer to as the bottleneck. With this design, we train the bottleneck to perform goal-conditioned future prediction, so that the latent representations encode future behaviors toward the desired goal, which are subsequently used to guide policy learning.

In the first stage (as shown in Figure 1), we use a bi-directional Transformer as TrajNet. The inputs include $T$ state tokens, one goal token, and a few learnable slot tokens. The action tokens are ignored. We first embed the goal token and all state tokens with separate linear encoders, then apply sinusoidal positional encoding before all inputs are sent into the Transformer Encoder.

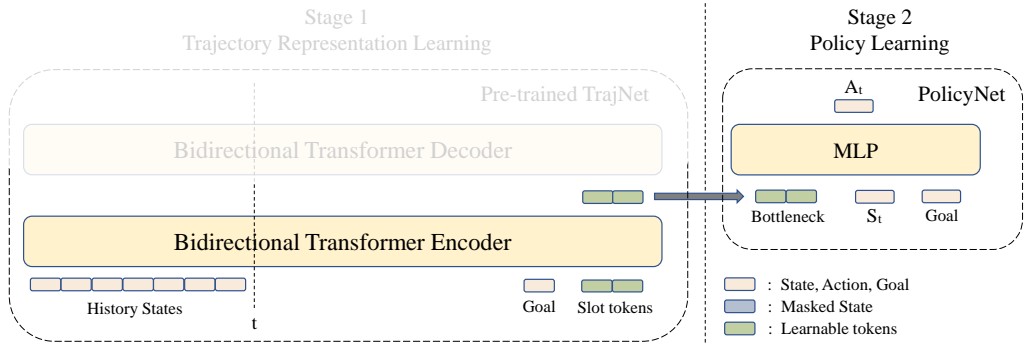

Figure 2: **Stage 2 – Policy Learning.** We implement policy learning with a simple MLP as PolicyNet. We input unmasked history states and retrieve the bottleneck from pre-trained encoder. Then the bottleneck is taken as the input to the policy network. We view the bottleneck generated by pre-trained encoder as goal-conditioned latent representations for the future trajectory.

While the full trajectories can be used for trajectory representation learning with TrajNet, only observed states are available during policy learning. For notation consistency, we denote the $k$ observed states as "history", and the $p$ states that follow the observed states as "future". The total number of input states for stage 1 is $T = k + p$, and for stage 2 is $k$. Both $k$ and $p$ are hyperparameters that represent history window length and future window length, respectively.

To perform goal-conditioned predictive coding, the entire future trajectory is masked. The tokens in the observed history are randomly masked (Figure 1 "MAE-RC"). The bottleneck is taken as the encoded slot tokens, which condenses the masked input trajectory into a compact representation. The decoder takes the bottleneck and $T$ masked tokens as the input, and aims to reconstruct both the history and the future states. Our intuition is that this will encourage the bottleneck to learn representations encoding the future trajectory and provide useful signals for decision making. The training objective of the TrajNet is to minimize MSE loss between the reconstructed and the ground-truth trajectory.

Figure 2 illustrates the interface between the TrajNet and the PolicyNet. PolicyNet is implemented as a simple MLP network. The trained TrajNet has its weights frozen, and only the TrajNet encoder is used to compute the trajectory representation $B_{\mathrm{obs}}$ from $k$ unmasked history states. PolicyNet takes the current state $s_t$, goal $g$, and bottleneck $B_{\mathrm{obs}}$ as input, and outputs an action. The training objective of PolicyNet is to minimize MSE loss between the predicted and ground-truth actions.

**Discussion**. Decoupling trajectory representation learning and policy learning allows us to explore different model's inputs and sequence modeling objectives. For example, recent work [2, 39] observed that the action sequences could potentially be detrimental to the learned policy in some tasks. In GCPC, we employ state-only trajectories as input and ignore the actions. Different objectives (e.g. masking patterns in the masked autoencoding objective) also have an impact on what is encoded in the bottleneck. Here we introduce five sequence modeling objectives for trajectory representation learning (as shown in Table 1) and study their impacts on policy learning. When using "AE-H" or "MAE-H", the bottleneck is only motivated to summarize the history states. When using "MAE-F" or "MAE-RC", the bottleneck is asked to perform predictive coding, and thus encode the future state sequences to achieve the provided goal. By default, we adopt the "MAE-RC" objective in GCPC.

Table 1: Objectives for Trajectory Representation Learning

| | Input | | Reconstruct | |
|---|---|---|---|---|
| | History | Future | History | Future |
| AE-H | Unmasked | – | ✓ | |
| MAE-H | Randomly Masked | – | ✓ | |
| MAE-F | Unmasked | Fully Masked | ✓ | ✓ |
| MAE-RC | Randomly Masked | Fully Masked | ✓ | ✓ |
| MAE-ALL | Randomly Masked | Randomly Masked | ✓ | ✓ |

## 4 Experiments

In this section, we aim to answer the following questions with empirical experiments: (1) Does sequence modeling benefit reinforcement learning via supervised learning on trajectory data, and how? (2) Is it beneficial to decouple trajectory representation learning and policy learning with bottlenecks? (3) What are the most effective trajectory representation learning objectives?

### 4.1 Experimental Setup

To answer the questions above, we conduct extensive experiments on three domains from D4RL offline benchmark suite [16]: AntMaze, FrankaKitchen and Gym Locomotion. AntMaze is a class of long-horizon navigation tasks, featuring partial observability, sparse reward and datasets that consist primarily of suboptimal trajectories. In this domain, an 8-DoF Ant robot needs to "stitch" parts of subtrajectories and navigates to a particular goal location in partially observed mazes. In addition to three mazes from the original D4RL, we also include a larger maze (AntMaze-Ultra) proposed by [28]. Both large and ultra setup in AntMaze poses significant challenges due to the complex maze

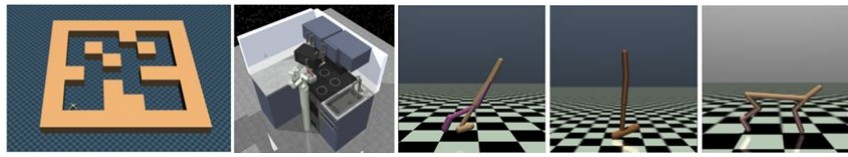

Figure 3: D4RL environments used in evaluation. From left to right: AntMaze, FrankaKitchen, Walker 2D, Hopper, and Halfcheetah.

layout and long navigation horizon. FrankaKitchen is a long-horizon manipulation task, in which a 9-DoF Franka robot arm is required to perform 4 subtasks in a simulated kitchen environment (e.g. open the microwave, turn on the light). In Gym Locomotion, we evaluate our approach on three continuous control tasks: Walker 2D, Hopper and Halfcheetah. Following [15], we refer to the "goal" of AntMaze and Kitchen as the target state configuration, and that of Gym as the average return-to-go.

**Experimental details.** For all benchmarks, we use a two-layer transformer encoder and a one-layer transformer decoder as the TrajNet, and a two-layer MLP as the PolicyNet (see detailed hyperparamters in A.3). During policy learning, we use the pre-trained TrajNet that achieves the lowest validation reconstruction loss to generate the bottleneck. For each evaluation run, we follow [29] to take the success rate over 100 evaluation trajectories for AntMaze tasks. For Kitchen and Gym Locomotion, we average returns over 50 and 10 evaluation trajectories, respectively. To determine the performance with a given random seed, we take the best evaluation result among the last five checkpoints (see discussion in A.2). For performance aggregation, we report the mean performance and standard deviation averaged over five seeds for each experiment.

### 4.2 Impact of Trajectory Representation Pretraining Objectives

In the two-stage framework, the pre-trained TrajNet generates trajectory representations in the form of the bottleneck, which is taken as an input to PolicyNet. To study the impact of trajectory representation learning objectives on the resulting policy performance, we implement five different sequence modeling objectives (as in Table 1) by varying masking patterns in the first stage pretraining.

Table 2 compares the performance of policies with different settings and pretraining objectives[1]. We note that MAE-F is the only effective masking pattern to perform zero-shot inference. After decoupling representation learning and policy learning, the MLP policy consistently outperforms the zero-shot transformer policy. This suggests good objectives for two stages could be different – by decoupling we can get the best of both worlds. Also, we observe that removing action sequences from trajectory representation pretraining yields performance gains in this task, whereas previous single-stage Transformer policy (e.g. [10]) usually requires action inputs to function, which further demonstrates the flexibility of our decoupled framework. With properly designed objectives, sequence modeling can generate powerful trajectory representations that facilitate the acquisition of performant policies.

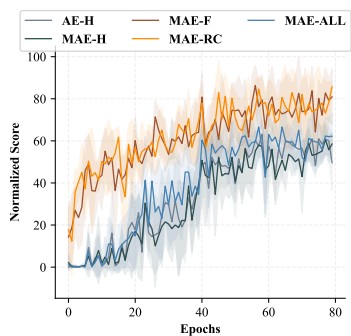

Figure 4: Policy learning curves with different pretraining objectives on Kitchen-Partial.

In both Table 2 and Figure 4, we observe that both MAE-F and MAE-RC outperform the other objectives in both AntMaze-Large and Kitchen-Partial, confirming the importance of the predictive coding objective for trajectory representation learning.

Table 2: **Comparison of trajectory representation pretraining objectives**. We evaluate five different objectives under three settings on large AntMaze environment: (1) Zero-shot: When actions are considered as part of the masked tokens, the pretrained TrajNet can be directly utilized as the policy; (2) Two-stage (w/o actions): Two-stage framework employs an MLP as PolicyNet, with state-only trajectories as the input to TrajNet. (3) Two-stage (w/ actions): Two-stage framework employs an MLP as PolicyNet, with state-action trajectories as the input to TrajNet.

| Large-Play | AE-H | MAE-H | MAE-F | MAE-RC | MAE-ALL |
|---|---|---|---|---|---|
| Zero-shot | - | 0 | $21.2 \pm 17.1$ | 0 | 0 |
| Two-stage (w/ actions) | $12.6 \pm 4.7$ | $6.4 \pm 3.8$ | $57.2 \pm 5.5$ | $62.0 \pm 10.7$ | $11.6 \pm 6.3$ |
| Two-stage (w/o actions) | $30.2 \pm 6.6$ | $36.6 \pm 12.6$ | $\mathbf{76.2 \pm 4.0}$ | $\mathbf{78.2 \pm 3.2}$ | $32.0 \pm 13.2$ |

### 4.3 The Role of Goal Conditioning in Trajectory Representation Pretraining

We also investigate whether goal conditioning (i.e. the goal input) in TrajNet is necessary or beneficial for learning trajectory representations. In Table 3, we observe that the objectives without performing

---

[1]For pre-training on AntMaze, when actions (i.e. the torque applied on joints) are considered, the full state space is reconstructed; when state-only trajectories are used, we reconstruct only the locations of the agent.

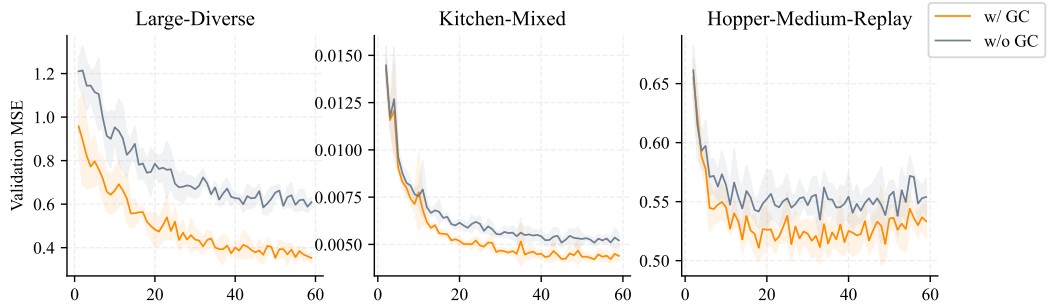

Figure 5: Reconstruction loss on the validation set during the trajectory representation learning stage, when MAE-F objective is used. "GC" refers to goal conditioning in TrajNet.

future prediction is insensitive to the goal conditioning. For example, the results of AE-H and MAE-H pretraining objectives align with the intuition that merely summarizing the history should not require the goal information. However, goal conditioning is crucial for predictive coding objectives (e.g. MAE-F and MAE-RC). Removing the goal from TrajNet might result in aimless prediction and misleading future representations, which thus harm the policy. Figure 5 shows the curves of validation loss during pretraining with MAE-F objective. We observe that goal conditioning reduces the prediction error and helps the bottleneck properly encode the expected long-term future, which would benefit the subsequent policy learning. One hypothesis is that specifying the goal enables the bottleneck to perform goal-conditioned implicit planning. The planned future may provide the correct waypoints for the agent to reach the distant goal. Figure 6 illustrates a qualitative result of the latent future in AntMaze, which depicts the future states decoded from the bottleneck using the pre-trained Transformer decoder. It demonstrates that the latent future with goal conditioning helps point out the correct direction towards the target location.

Table 3: **Ablation study of goal conditioning on AntMaze-Large**. Removing the goal conditioning from TrajNet would seriously affect predictive coding objectives and harm the resulting policy performance, suggesting that the properly encoded future representation provides crucial guidance for performing long-horizon tasks. "GC" refers to goal conditioning in TrajNet.

| Large-Play | AE-H | MAE-H | MAE-F | MAE-RC | MAE-ALL |
|---|---|---|---|---|---|
| w/o GC | $32.8 \pm 4.3$ | $33.2 \pm 11.9$ | $10.0 \pm 2.2$ | $16.0 \pm 5.3$ | $36.2 \pm 10.4$ |
| w/ GC | $30.2 \pm 6.6$ | $36.6 \pm 12.6$ | $\mathbf{76.2} \pm \mathbf{4.0}$ | $\mathbf{78.2} \pm \mathbf{3.2}$ | $32.0 \pm 13.2$ |

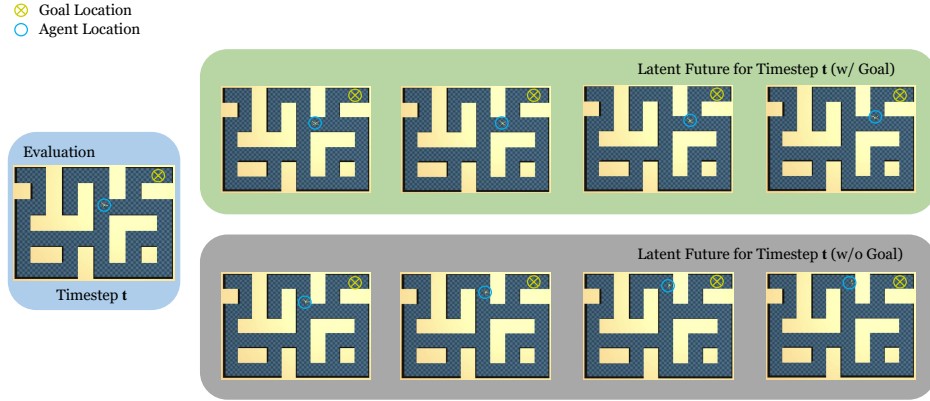

Figure 6: **Latent future visualization on AntMaze-Large.** A qualitative result comparing the latent future with and without goal conditioning. Given the goal information, the bottleneck can encode the latent future moving in the desired direction.

## 4.4 Latent Future versus Explicit Future as Policy Conditioning Variables

Prior work [27, 26] has demonstrated that planning into the future is helpful for solving long-horizon tasks. These work performs planning on the explicit future by sampling desirable future states (or transitions), while GCPC leverages goal-conditioned latent representations that encode the future state sequence. In this experiment, we examine how the implicit encoding of future information affects policy performance when it serves as a conditioning variable of PolicyNet. Specifically, we obtain the bottleneck from a TrajNet, which is pre-trained with $p$-length future window. The bottleneck encodes latent representations of $p$ future states. Correspondingly, we acquire the same number of explicit future states using the pre-trained transformer decoder (see Appendix A.4). Either the bottleneck or explicit future states are taken as auxiliary inputs to the PolicyNet. In Table 4, we evaluate the policy with $p = 70$ in large AntMaze and $p = 30$ in Kitchen. The results illustrate that the bottleneck is a powerful future information carrier that effectively improves the policy performance. Different from previous approaches that estimate explicit future states with step-by-step dynamics models [27], using latent future representations can mitigate compounding rollout errors. Compared to diffusion-based planning methods [2], which require iterative refinement to obtain a future sequence, latent future avoids the time-consuming denoising step and reduces the decision latency.

Table 4: **Comparison between latent and explicit future**. Compared to explicit future states, the latent future encoded by the bottleneck is more effective for policy learning. $p$ is future window size.

|  | Large-Play | Large-Diverse | Kitchen-Mixed | Kitchen-Partial |
|---|---|---|---|---|
|  | $p = 70$ | | $p = 30$ | |
| Explicit Future | $67.9 \pm 9.3$ | $70.0 \pm 4.6$ | $72.4 \pm 4.5$ | $73.9 \pm 9.3$ |
| Latent Future | $\mathbf{78.2} \pm \mathbf{3.2}$ | $\mathbf{80.6} \pm \mathbf{3.9}$ | $\mathbf{75.6} \pm \mathbf{0.8}$ | $\mathbf{90.2} \pm \mathbf{6.6}$ |

Table 5: Average normalized scores of GCPC against other baselines on **AntMaze**. TAP's performances are taken from the original paper [28]. For other baselines, we obtain their performance by re-running author-provided or our replicated implementations with our evaluation protocol. Following [29], we bold all scores within 5 percent of the maximum per task ($\geq 0.95 \cdot$ max).

| Dataset | BC | CQL | IQL | DT | TAP | WGCSL | GCIQL | DWSL | RvS-G | GCPC |
|---|---|---|---|---|---|---|---|---|---|---|
| Umaze | $63.4 \pm 9.4$ | $\mathbf{88.2} \pm \mathbf{2.3}$ | $\mathbf{92.8} \pm \mathbf{3.4}$ | $55.6 \pm 6.3$ | - | $\mathbf{90.8} \pm \mathbf{2.8}$ | $\mathbf{91.6} \pm \mathbf{4.0}$ | $71.2 \pm 4.2$ | $70.4 \pm 4.0$ | $71.2 \pm 1.3$ |
| Umaze-Diverse | $63.4 \pm 4.4$ | $47.4 \pm 2.0$ | $71.2 \pm 7.0$ | $53.4 \pm 8.6$ | - | $55.6 \pm 15.7$ | $\mathbf{88.8} \pm \mathbf{2.2}$ | $74.6 \pm 2.8$ | $66.2 \pm 5.6$ | $71.2 \pm 6.6$ |
| Medium-Play | $0.6 \pm 0.5$ | $72.8 \pm 5.7$ | $75.8 \pm 1.3$ | 0 | 78.0 | $63.2 \pm 13.7$ | $\mathbf{82.6} \pm \mathbf{5.4}$ | $77.6 \pm 3.0$ | $71.8 \pm 4.7$ | $70.8 \pm 3.4$ |
| Medium-Diverse | $0.4 \pm 0.5$ | $70.8 \pm 10.3$ | $76.6 \pm 4.2$ | 0 | **85.0** | $46.0 \pm 12.6$ | $76.2 \pm 6.3$ | $74.8 \pm 9.3$ | $72.0 \pm 3.7$ | $72.2 \pm 3.4$ |
| Large-Play | 0 | $36.4 \pm 10.3$ | $50.0 \pm 9.7$ | 0 | 74.0 | $0.6 \pm 1.3$ | $40.0 \pm 16.2$ | $15.2 \pm 7.7$ | $35.6 \pm 7.6$ | $\mathbf{78.2} \pm \mathbf{3.2}$ |
| Large-Diverse | 0 | $36.0 \pm 8.3$ | $52.6 \pm 5.9$ | 0 | **82.0** | $2.4 \pm 4.3$ | $29.8 \pm 6.8$ | $19.0 \pm 2.8$ | $25.2 \pm 4.8$ | $\mathbf{80.6} \pm \mathbf{3.9}$ |
| Ultra-Play | 0 | $18.0 \pm 13.3$ | $21.2 \pm 7.5$ | 0 | 22.0 | $0.2 \pm 0.4$ | $20.6 \pm 7.6$ | $25.2 \pm 3.0$ | $25.6 \pm 6.7$ | $\mathbf{56.6} \pm \mathbf{9.5}$ |
| Ultra-Diverse | 0 | $9.6 \pm 14.6$ | $17.8 \pm 4.0$ | 0 | 26.0 | 0 | $28.4 \pm 11.8$ | $25.0 \pm 8.6$ | $26.4 \pm 7.7$ | $\mathbf{54.6} \pm \mathbf{10.3}$ |
| Average | 16.0 | 47.4 | 57.3 | 13.6 | - | 32.4 | 57.3 | 47.8 | 49.2 | **69.4** |

## 4.5 Effectiveness of GCPC

Finally, we show the effectiveness of GCPC by evaluating it on three different domains: AntMaze, Kitchen and Gym Locomotion.

**Baselines and prior methods.** We compare our approach to both supervised learning methods and value-based RL methods. For the former, we consider: (1) Behavioral Cloning (BC), (2) RvS-R/G [15], a conditional imitation learning method that is conditioned on either a target state or an expected return. (3) Decision Transformer (DT) [10], a return-conditioned model-free method that learns a transformer-based policy, (3) Trajectory Transformer (TT) [27], a model-based method that performs beam search on a transformer-based trajectory model for planning, and (4) Decision Diffuser (DD) [2], a diffusion-based planning method that synthesizes future states with a diffusion model and acts by computing the inverse dynamics. For the latter, we select methods based on dynamic programming, including (5) CQL [31] and (6) IQL [29]. Additionally, we include three goal (state)-conditioned baselines for AntMaze and Kitchen: (7) Goal-conditioned IQL (GCIQL), (8) WGCSL [52] and (9) DWSL [25]. AntMaze-Ultra is a customized environment proposed by [28], therefore we include TAP's performance on the v0 version of AntMaze for completeness. When feasible, we re-run the baselines with our evaluation protocol for fair comparison [1].

Our empirical results on AntMaze and Kitchen are presented in Table 5 and Table 6. We find our methods outperform all previous methods on the average performance. In particular, on the most challenging large and ultra AntMaze environments, our method achieves significant improvements over the RvS-G baseline, demonstrating the efficacy of learning good future representations using sequence modeling in long-horizon tasks. Table 7 shows the results on Gym Locomotion tasks. Our approach obtains competitive performance as prior methods. We also notice that compared to Decision Transformer, RvS-R can already achieve strong average performance. This suggests that for some tasks sequence modeling may not be a necessary component for policy improvement. With a large fraction of near-optimal trajectories in the dataset, a simple MLP policy may provide enough capacity to handle most of the locomotion tasks.

Table 6: Average normalized scores of GCPC against other baselines on **Kitchen**. CQL and DD's results are taken from [2]. For other baselines, we obtain their performance by re-running author-provided or our replicated implementations with our evaluation protocol. Following [29], we bold all scores within 5 percent of the maximum per task ($\geq 0.95 \cdot$ max).

| Dataset | BC | CQL | IQL | DT | DD | WGCSL | GCIQL | DWSL | RvS-G | GCPC |
|---|---|---|---|---|---|---|---|---|---|---|
| Mixed | $48.9 \pm 0.7$ | 52.4 | $53.2 \pm 1.6$ | $50.7 \pm 7.1$ | 65.0 | $\mathbf{77.8} \pm 3.6$ | $\mathbf{74.6} \pm 1.9$ | $\mathbf{74.6} \pm 0.6$ | $69.4 \pm 4.2$ | $\mathbf{75.6} \pm 0.8$ |
| Partial | $41.3 \pm 3.7$ | 50.1 | $59.7 \pm 8.3$ | $48.6 \pm 9.5$ | 57.0 | $75.2 \pm 6.4$ | $74.7 \pm 4.1$ | $74.0 \pm 5.8$ | $71.7 \pm 7.9$ | $\mathbf{90.2} \pm 6.6$ |
| Average | 45.1 | 51.3 | 56.5 | 49.7 | 61.0 | 76.5 | 74.7 | 74.3 | 70.6 | **82.9** |

Table 7: Average normalized scores of our approach against other baselines on **Gym Locomotion**. TT and DD's results are taken from [2]. For other baselines, we obtain their performance by re-running author-provided or our replicated implementations with our evaluation protocol. Following [29], we bold all scores within 5 percent of the maximum per task ($\geq 0.95 \cdot$ max).

| Dataset | Environment | BC | CQL | IQL | DT | TT | DD | RvS-R | GCPC |
|---|---|---|---|---|---|---|---|---|---|
| Medium-Expert | HalfCheetah | $61.9 \pm 5.8$ | $87.7 \pm 7.0$ | $\mathbf{92.4} \pm 0.4$ | $88.8 \pm 2.6$ | 95 | 90.6 | $\mathbf{93.4} \pm 0.3$ | $\mathbf{94.0} \pm 0.9$ |
| Medium-Expert | Hopper | $55.1 \pm 2.8$ | $\mathbf{110.5} \pm 2.9$ | $104.2 \pm 8.7$ | $\mathbf{108.4} \pm 2.0$ | **110** | 111.8 | $\mathbf{111.3} \pm 0.2$ | $\mathbf{111.7} \pm 0.4$ |
| Medium-Expert | Walker2d | $100.4 \pm 13.4$ | $\mathbf{110.4} \pm 0.6$ | $\mathbf{110.2} \pm 0.7$ | $108.6 \pm 0.3$ | 101.9 | 108.8 | $\mathbf{109.6} \pm 0.4$ | $\mathbf{109.0} \pm 0.2$ |
| Medium | HalfCheetah | $43.0 \pm 0.4$ | $\mathbf{47.1} \pm 0.3$ | $\mathbf{47.7} \pm 0.2$ | $42.9 \pm 0.2$ | 46.9 | **49.1** | $44.2 \pm 0.2$ | $44.5 \pm 0.5$ |
| Medium | Hopper | $55.8 \pm 2.8$ | $70.1 \pm 1.6$ | $69.2 \pm 3.2$ | $67.8 \pm 4.2$ | 61.1 | **79.3** | $65.1 \pm 5.2$ | $68.0 \pm 4.4$ |
| Medium | Walker2d | $74.1 \pm 3.2$ | $\mathbf{83.5} \pm 0.5$ | $\mathbf{84.5} \pm 1.5$ | $76.5 \pm 1.6$ | 79 | 82.5 | $78.4 \pm 2.6$ | $78.0 \pm 2.4$ |
| Medium-Replay | HalfCheetah | $37.2 \pm 1.3$ | $\mathbf{45.4} \pm 0.3$ | $44.9 \pm 0.3$ | $37.8 \pm 0.9$ | 41.9 | 39.3 | $40.2 \pm 0.2$ | $40.7 \pm 1.5$ |
| Medium-Replay | Hopper | $33.7 \pm 8.5$ | $\mathbf{96.2} \pm 1.9$ | $93.9 \pm 9.1$ | $78.0 \pm 11.6$ | 91.5 | **100** | $88.5 \pm 12.9$ | $94.2 \pm 3.5$ |
| Medium-Replay | Walker2d | $19.2 \pm 7.3$ | $\mathbf{79.8} \pm 1.6$ | $78.6 \pm 5.7$ | $72.5 \pm 3.3$ | **82.6** | 75 | $71.0 \pm 5.1$ | $77.6 \pm 9.8$ |
| Average | | 53.4 | **81.2** | 80.6 | 75.7 | **78.9** | 81.8 | 78.0 | **79.7** |

## 5 Conclusion

In this work, we aim to investigate the role of sequence modeling in learning trajectory representations and its utility in acquiring performant policies. To accomplish this, we employ a two-stage framework that decouples trajectory representation learning and policy learning. This framework unifies many existing RvS methods, enabling us to study the impacts of different trajectory representation learning objectives for sequential decision making. Within this framework, we introduce a specific design – Goal-conditioned Predictive Coding (GCPC), that incorporates a compact bottleneck to transfer representations between the two stages and learns goal-conditioned latent representations modeling the future trajectory. Through extensive experiments, we observe that the bottleneck generated by GCPC properly encodes the goal-conditioned future information and brings significant improvement over some long-horizon tasks. By empirically evaluating our approach on three benchmarks, we demonstrate GCPC achieves competitive performance on a variety of tasks.

**Limitations and Future work.** GCPC models the future by performing maximum likelihood estimation on offline collected trajectories, which may predict overly optimistic future behaviors and lead to suboptimal actions in stochastic environments. Future work includes discovering policies that are robust to the environment stochasticity by considering multiple possible futures generated by GCPC. Another limitation of our work is that GCPC may not be sufficient to maintain high accuracy for long-term future prediction when high-dimensional states are involved, which may potentially be tackled by leveraging foundation models to acquire representations for high-dimensional inputs.

## Acknowledgments and Disclosure of Funding

We appreciate all anonymous reviewers for their constructive feedback. We would like to thank Calvin Luo and Haotian Fu for their discussions and insights, and Tian Yun for the help on this project. This work is in part supported by Adobe, Honda Research Institute, Meta AI, Samsung Advanced Institute of Technology, and a Richard B. Salomon Faculty Research Award for C.S.

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

# A  Experimental Details

## A.1  Additional Baseline Details

We provide additional information on the baseline methods and their performance we present in the main paper. We compare our proposed method with (1) supervised learning methods, including Behavioral Cloning (BC), RvS, Decision Transformer (DT), Trajectory Transformer (TT), and Decision Diffuser (DD). We also compare with (2) value-based methods, including Conservative Q-Learning (CQL) and Implicit Q-Learning (IQL). DT uses a transformer encoder to model offline-RL as an autoregressive sequence generation problem in a model-free way, which predicts future actions based on previous states, actions, and returns-to-go. Similarly, The model-based TT also adopts transformer structure for sequence prediction but uses beam search for planning during execution. Inspired by conditional generative model, DD adopts a diffusion model to sample future state sequence based on returns or constraints and extract actions between the states by using a inverse dynamics model with relyng on value estimation. Similar to our PolicyNet, RvS-G and RvS-R take the current state and the goal (return or state) as input, and train a two-layer MLP to predict the actions via supervised learning. For value-based methods, in order to solve the problem of overestimating values induced by distributional shift, CQL adds value regularization terms to the standard Bellman error objective to learn a lower bound of the true Q-function. IQL approximates the policy improvement step implicitly instead of updating the Q-function with target actions sampled from the behavior policy. For the additional goal (state)-conditioned baselines, GCIQL is a goal-conditioned version of IQL. WGCSL enhances GCSL with an advanced compound weight, which optimizes the lower-bound of the goal-conditioned RL objective. DWSL first models the distance between states and the goal, and extracts the policy by imitating the actions that reduce the minimum distance metric.

**Baseline Performance Source**. For AntMaze, we use v2 version for D4RL environments and v0 version for the additional AntMaze-Ultra environments. For Gym Locomotion and Kitchen, we use v2 and v0 version of the datasets from the D4RL benchmark, respectively. For all baselines except TT, DD and TAP, we obtain their performance by re-running author-provided (CQL, IQL, DT, WGCSL, GCIQL, DWSL) or our replicated (BC, RvS-G/R) implementations with our evaluation protocol. For CQL's performance on Kitchen, we are unable to reproduce the previously reported performance with the default hyperparameters, so we take the corresponding results from [2]. For additional goal (state)-conditioned baselines (i.e. WGCSL, GCIQL and DWSL), we take the implementations provided by DWSL authors[1], and use the same goal specification (see details in A.6) for AntMaze and Kitchen as ours. For TT, DD and TAP's performance, we take the results from the original papers.

## A.2  Evaluation Protocol

Many prior work [48, 18] reports their performance by taking the final checkpoint's evaluation results and averaging them over multiple seeds. However, in some cases we observe that this evaluation protocol can lead to large oscillations in the policy performance when gradient steps (or training epochs) are slightly perturbed or the random seeds are changed (as in Figure A1). As different methods may use various number of training steps, seeds and evaluation trajectories, it hampers drawing robust conclusions by comparing these results. To faithfully capture the performance trend, which is crucial for ablation analysis, for each seed we take the best evaluation result among the last five checkpoints as its performance, where each evaluation result is the average return over $N$ evaluation trajectories (e.g. $N = 10, 50, 100$) achieved by a given checkpoint, and then we calculate the mean performance and standard deviation over five seeds. Empirically we find this evaluation protocol yields results that are less sensitive to the above factors and usually have smaller error bounds, facilitating the comparisons in ablation studies. Meanwhile, we observe that in some cases this evaluation protocol might lead to results that are higher than those reported in previous papers, especially when the policy has more drastic performance fluctuations during training. We re-run most baselines using the same evaluation protocol for fair comparisons.

To investigate the impact of different aggregation metrics, we follow [1] to compare performance aggregation using mean, median, IQM and optimality gap on the AntMaze tasks. In Figure A2, we observe that GCPC consistently outperforms compared baselines across all metrics.

---

[1]https://github.com/jhejna/dwsl/

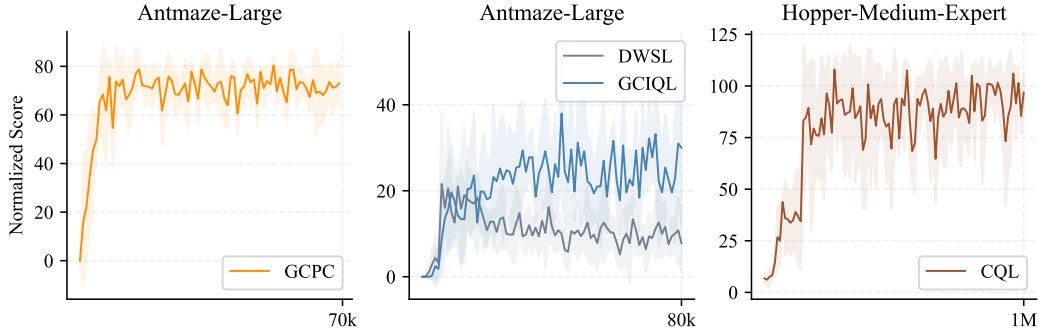

Figure A1: Training curves of different methods with various default gradient steps

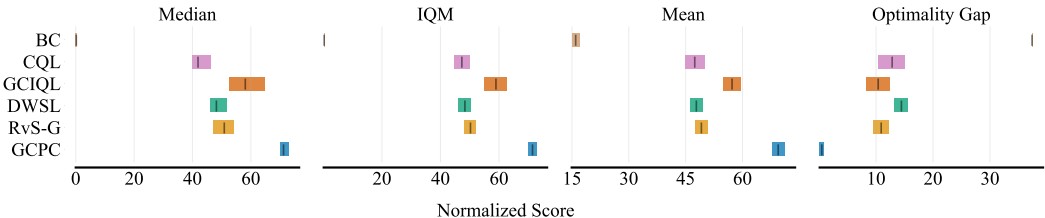

Figure A2: Different aggregation metrics with 95% confidence intervals on AntMaze tasks.

Apart from model-free policies, it also remains unclear whether or how the results of planning-based methods [2, 27] would be affected by the choice of evaluation protocol. We leave adopting more reliable and robust evaluation approaches for comprehensive analysis to the future work.

### A.3 Hyperparameter

We list hyperparameters for BC and RvS-G/R replication in Table A1 and GCPC implementation in Table A2. For RvS-G/R and PolicyNet in GCPC, we use a two-layer feedforward MLP as the policy network, taking the current state and goal (state or return-to-go) as input, the only difference is that PolicyNet takes the bottleneck as an additional input. All experiments are performed on a single Nvidia RTX A5000.

Table A1: Replicated BC and RvS-G/R Hyperparameter

| Hyperparameters | Value |
| --- | --- |
| Hidden Layers | 2 |
| Embedding Dim | 1024 |
| Nonlinearity | ReLU |
| Epochs | 80 AntMaze, Kitchen |
|  | 100 Gym Locomotion |
| Batch Size | 1024 AntMaze, Gym Locomotion |
|  | 256 Kitchen |
| Dropout | 0 |
| Optimizer | Adam |
| Learning Rate | 1e-3 |

### A.4 Explicit future

Existing methods, like TT and DD, utilize future information by sampling future states (or transitions) and planning on the explicit future. TT learns a world model that autoregressively generates states, actions, rewards and return-to-go estimates, then uses beam search to select future trajectories

Table A2: GCPC Hyperparameter

| | Hyperparameters | Values |
|---|---|---|
| | Optimizer | Adam |
| TrajNet (Bidirectional Transformer) | Encoder Layers | 2 |
| | Decoder Layers | 1 |
| | Attention Heads | 4 |
| | Embedding Dim | 256 |
| | Nonlinearity | GELU |
| | Slot Tokens | 4 |
| | (History Window $k$, Future Window $p$) | (10, 70) AntMaze |
| | | (5, 30) Kitchen |
| | | (5, 20) Locomotion |
| | Epochs | 60 AntMaze, Locomotion |
| | | 20 Kitchen |
| | Batch Size | 1024 |
| | Dropout | 0.1 |
| | Learning Rate | 1e-4 |
| PolicyNet (Multi-Layer Perceptron) | Hidden Layers | 2 |
| | Embedding Dim | 1024 |
| | Nonlinearity | ReLU |
| | Epochs | 80 AntMaze, Kitchen |
| | | 100 Gym Locomotion |
| | Batch Size | 1024 AntMaze, Gym Locomotion |
| | | 256 Kitchen |
| | Dropout | 0 |
| | Learning Rate | 1e-3 AntMaze, Gym Locomotion |
| | | 1e-4 Kitchen |

with highest cumulative rewards. DD leverages the diffusion process with classifier-free guidance to generate future states that satisfy reward maximization or other constraints, and uses inverse dynamics to extract actions between future states.

In GCPC, we compress the goal-conditioned future into latent representations and use them to support policy learning. In Section 4.4, we study the effect of explicitly leveraging the decoded future states. We adapt our framework as follows: During the policy learning and evaluation, we first freeze transformer encoder and decoder trained in the first stage. Instead of directly passing the encoded bottlenecks to the PolicyNet, we leverage the pre-trained transformer decoder to decode the bottleneck into goal-conditioned futures. The new PolicyNet is modified to take the observed states, the goal, and the concatenation of the decoded future states. The bottleneck is discarded and not used by the modified PolicyNet.

## A.5  Future window length

We further investigate the effect of future window length on policy performance, which decides how far the bottleneck can look into the future during the first stage pretraining. In Table A3, we show that predicting further future would generally bring bigger improvements in AntMaze-Large, emphasizing that planning into long-term future can largely benefit policy performance. We also observe that the improvements brought by enlarging future window are specific to both datasets and the range of future window length. The improvements become marginal once the future window expands beyond a certain size.

## A.6  Goal Sampling and Specification

To perform goal-conditioned imitation learning, we follow [15] to sample an outcome from the future trajectory $\tau_{t:H}$ as the goal. When the goal is in the form of a target state (e.g. in AntMaze and Kitchen tasks), we randomly sample a reachable future state from $\{s_i\}_{i=t+1}^{H}$ and keeps the task-related subspace as $g$. Specifically, for AntMaze we take the ant's $(x, y)$ location as the goal, which are the first two dimensions of the state space; for Kitchen's goal, we keep the dimensions

Table A3: **Future window length**. In AntMaze-Large, the farther into the future can the bottleneck see during trajectory representation pre-training, the more helpful they are for policy learning.

| Future window length $p$ | 20 | 40 | 70 |
|---|---|---|---|
| Large-Diverse | $62.4 \pm 5.6$ | $\mathbf{78.2} \pm 4.8$ | $\mathbf{80.6} \pm 3.9$ |
| Large-Play | $49.6 \pm 6.7$ | $\mathbf{76.6} \pm 5.0$ | $\mathbf{78.2} \pm 3.2$ |
| Average | 56.0 | **77.4** | **79.4** |

that indicate the position of the target object in all 7 subtasks and zero out other dimensions (e.g. dimensions that indicate the robot proprioceptive state) in the 30-dimensional state space[2]. When the goal is represented as the expected return-to-go (e.g. in Gym Locomotion tasks), we use the average return achieved over some number of timesteps into the future as $g$ (i.e. $g = \frac{1}{H-t+1} \sum_{i=t}^{H} r_i$, where we follow [15] to set $H$ to the constant maximum episode length).

# B    Pseudocode of GCPC

---
**Algorithm 1** Goal-conditioned Predictive Coding (GCPC) for RvS
---
1: **Input**: Dataset of trajectories $\mathcal{D} = \{\tau\}$, History window length $k$, Future window length $p$, Masking ratio $r$, Bi-directional transformer encoder $h_{\text{enc}}$, Bi-directional transformer decoder $h_{\text{dec}}$, Learnable slot tokens $Q$, Policy network $\pi$
2: **while** First Stage **do**
3:      Sample a trajectory and a timestep: $\tau = \{(s_i, a_i)\}_{i=1}^{H} \sim \mathcal{D}, t \sim [k, H]$
4:      Sample a goal $g$, obtain observed states and full states: $s_h = s_{t-k+1:t}$, $s_w = s_{t-k+1:t+p}$
5:      Randomly mask observed states with masking ratio $r$: $s_h^m = \texttt{Masked}(s_h, r)$
6:      Encode masked observed states, goal and learnable slot tokens: $B = h_{\text{enc}}(Q, s_h^m, g)$
7:      Reconstruct observed and future states with the bottleneck $B$: $\hat{s}_w = h_{\text{dec}}(B)$
8:      Compute loss: $\mathcal{L} = \texttt{MSE}(\hat{s}_w, s_w)$
9:      Update parameters in $h_{\text{enc}}$ and $h_{\text{dec}}$
10: **end while**
11: Freeze parameters in $h_{\text{enc}}$
12: **while** Second Stage **do**
13:      Sample a trajectory and a timestep: $\tau = \{(s_i, a_i)\}_{i=1}^{H} \sim \mathcal{D}, t \sim [k, H]$
14:      Sample a goal $g$, obtain observed states: $s_h = s_{t-k+1:t}$
15:      Encode observed states, goal and learnable slot tokens: $B_{\text{obs}} = h_{\text{enc}}(Q, s_h, g)$
16:      Train the policy network $\pi$ with the bottleneck $B_{\text{obs}}$: $\hat{a}_t = \pi(B_{\text{obs}}, s_t, g)$
17:      Compute loss: $\mathcal{L} = \texttt{MSE}(\hat{a}_t, a_t)$
18:      Update parameters in $\pi$
19: **end while**
20: **return** Policy network $\pi$
---

# C    Pre-training mask ratio

In the first stage of GCPC, we randomly mask history states with a masking ratio $r$. We compare using fixed masking ratios (20%, 40%, 80%) and dynamic masking ratio "D" uniformly sampled from 0%, 20%, 40%, 60% and 80%. Figure A3 shows the influence of different masking ratios on AntMaze and FrankaKitchen. We found that low masking ratios (e.g. 20%) would usually perform worse except on the Kitchen-Mixed dataset, and a high masking ratio (80%) turns out to be more effective, which is similar to the observation in MAE [24]. However, the optimal masking ratio is not consistent across environments. In order to achieve the best overall performance, we adopt dynamic masking ratio "D" in our implementation.

---
[2]In the Kitchen experiments, we found it important to zero-out redundant dimensions, such as robot proprioceptive dimensions, when specifying the task goal for goal (state)-conditioned methods. We use this specification for all goal (state)-conditioned methods.

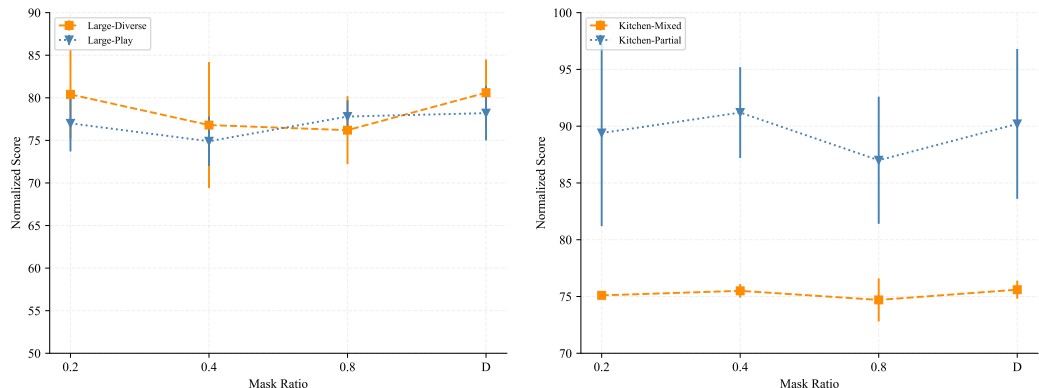

Figure A3: Mask ratio ablation on AntMaze and FrankaKitchen. "D" refers to dynamic mask ratio randomly drawn from 0%, 20%, 40%, 60% and 80%

## D    Additional Experiments on FrankaKitchen

In Table A4, we present how different pretraining objectives affect policy performance on the FrankaKitchen benchmark. MAE-F and MAE-RC are the only valid objectives in zero-shot setting. In two-stage setting, we observe that action inputs would generally cause larger variance and harm the performance of two predictive coding objectives. Similar to our observations on AntMaze, predictive coding objectives can consistently generate powerful trajectory representations, which yield performance gains on Kitchen tasks.

Table A4: **Objective Ablation on Kitchen-Partial**. We report mean and standard error over 5 seeds.

| Kitchen-Partial | AE-H | MAE-H | MAE-F | MAE-RC | MAE-ALL |
|---|---|---|---|---|---|
| Zero-shot | – | 0 | $4.4 \pm 9.8$ | $4.4 \pm 6.5$ | 0 |
| Two-stage (w/ actions) | $76.9 \pm 23.9$ | $73.6 \pm 16.6$ | $86.4 \pm 12.5$ | $80.4 \pm 18.0$ | $66.8 \pm 14.5$ |
| Two-stage (w/o actions) | $66.4 \pm 5.9$ | $72.7 \pm 7.0$ | $92.5 \pm 4.9$ | $90.2 \pm 6.6$ | $68.0 \pm 4.9$ |

## E    Visualizing the Latent Future

In this section, we visualize the latent future by decoding the bottlenecks back into the state space using the pre-trained decoder. We visualize the currently observed state and the decoded future states inside the maze environment. Figure A4 includes two examples extracted from successful evaluation rollouts. The blue part on the left shows the actual position of the agent at a certain timestep $s_t$ during the evaluation and the green part on the right is the latent future predicted based on history states $s_{\leq t}$ and the goal $g$. We find in the latent future, the agent gradually move towards the goal, which demonstrates that bottlenecks can encode meaningful future trajectories that provide the correct waypoints as subgoals and bring the agent from the current location to the final goal.

Figure A5 illustrates how latent future vary depending on different goal conditions for the same current state. By specifying different goals, the latent future would change correspondingly and point out the right direction for the agent. These visualizations suggest that the bottlenecks have the ability to provide the correct intermediate waypoints based on history states and the goal, which is crucial for long-horizon goal-reaching tasks. In the second example of Figure A5, multiple paths connect the agent and the goal, but bottlenecks encode the shortest one for the agent, which also demonstrates the bottleneck's ability to stitch sub-optimal trajectories and compose the optimal path.

In Section 4.3, we found that removing the goal from TrajNet input would seriously harm the performance on large AntMaze. In Figure A6, we visualize the latent future without goal conditioning. In both examples, bottlenecks encode latent futures that either move in the opposite direction to the goal or enter a dead end and cause tasks to fail. It again demonstrates that the quality of the encoded

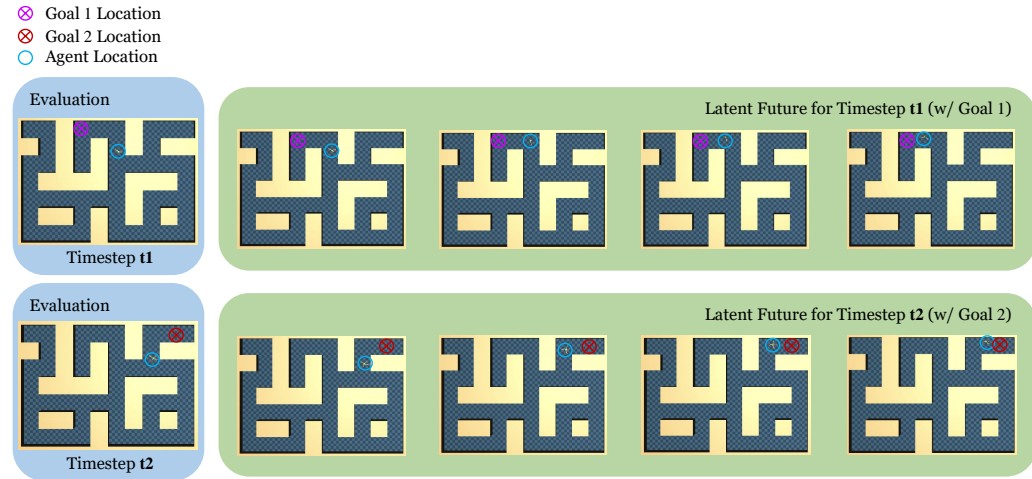

Figure A4: Two latent future examples from successful rollouts. Latent future shows the right path to the goal.

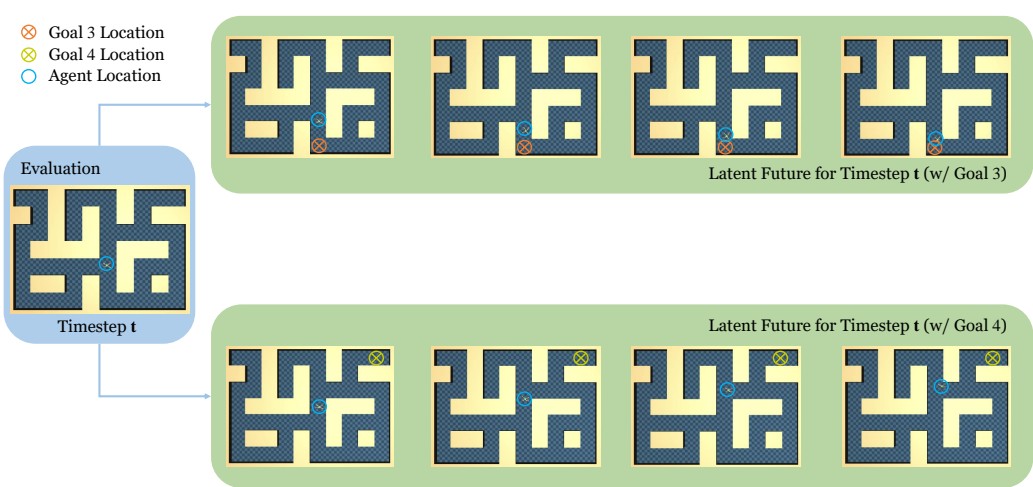

Figure A5: Latent future conditioned on different goals in large AntMaze.

future would significantly affect how policy performs, and providing goal information can prevent misleading future.

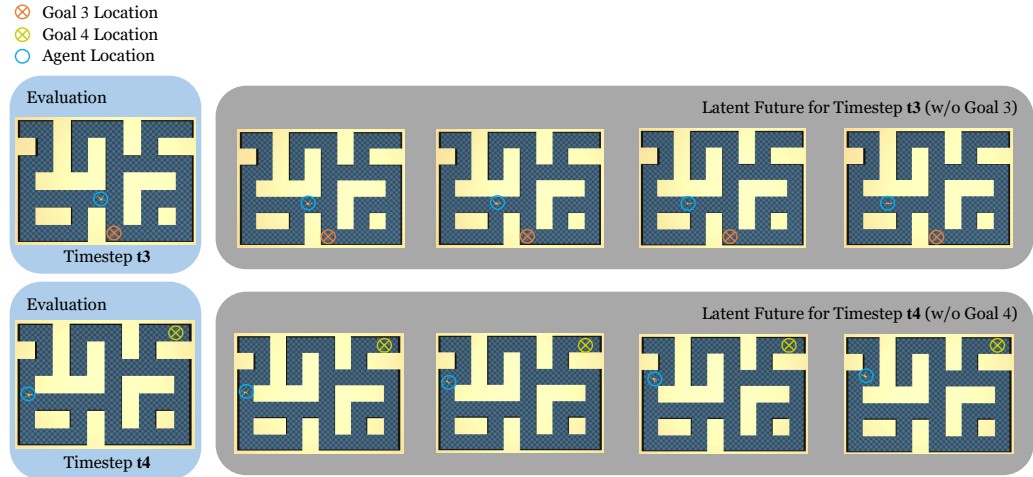

Figure A6: Failure cases after removing goal from TrajNet input. Latent future without goal conditioning would mislead the agent.

# F    Slot tokens ablation

Table A5 shows how changing the number of slot tokens would affect policy performance on large AntMaze. We observed that using different numbers of slot tokens would not have big impacts on the performance. In our implementation, we uniformly use 4 slot tokens for both trajectory representation learning and policy learning.

Table A5: **The number of slot tokens**. We report mean and standard error over 3 seeds.

| # Slot Tokens | 1 | 2 | 4 |
|---|---|---|---|
| Large-Diverse | $80.6 \pm 3.5$ | $77.2 \pm 4.9$ | $80.6 \pm 3.9$ |
| Large-Play | $81.2 \pm 4.4$ | $78.8 \pm 4.7$ | $78.2 \pm 3.2$ |

