# OpenReview forum: "Goal-Conditioned Predictive Coding for Offline Reinforcement Learning"
_NeurIPS.cc/2023/Conference — NeurIPS 2023 poster_

### Official Review · Reviewer_EZxU · 2023-06-25

**Soundness:** 4 excellent
**Presentation:** 2 fair
**Contribution:** 4 excellent
**Rating:** 6
**Confidence:** 4

**Summary:**

Update 8/24: As discussed with the authors, I'm updating my score to "weak accept."

-----------------------

This paper studies the role of representation learning for goal-conditioned behavioral cloning methods, focusing on those representations produced by sequence models. The paper fines that a statewise approach, first training a sequence model to produce representations and then using those representations as an _additional_ input to an MLP, outperforms end-to-end approaches. The paper also includes detailed ablation experiments studying how those representations should be learned (don't include actions, among other findings).

**Strengths:**

**Originality**: This paper brings a fresh perspective to a class of RL methods that are becoming increasingly popular. I appreciate that, rather than focus on proposing a new method (which the paper admittedly does), much of the focus is on understanding the importance of design decisions from prior work.

**Quality**:
* The experiments seem quite thorough. Rather than just aiming to show that the proposed method is better than baselines, the paper includes a thorough investigation of how the representations should be learned, and includes experiments on a range of standard benchmarks. The experimental design seems clever and useful for probing the design decisions.

**Clarity**:
* The abstract, introduction, and related work sections are well written.
* The introduction does a great job motivating the problem.
* Figures 1 and 2 are useful for providing an overview of the method. The transparency on the left of Fig 2 is a useful visual trick.

**Significance**
* I think the paper is studying a really interesting problem, trying to dissect exactly why sequence models can be effective for conditional behavioral cloning.

**Weaknesses:**

1. The main weakness of the paper is that the experiments section is unclear, making it difficult to understand the experiments. Some of the confusion is from the different masking strategies, which I didn't see explicitly defined anywhere. Evening spelling out the names would help  (I'm guessing that "AE" is autoencoder and "MAE" is masked autoencoder, but I'm unsure what the other parts mean).
2. I'm unsure about the claims about implicit planning. I admittedly didn't understand these experiments thoroughly (see #1). To claim that a method is performing planning, it seems like we'd need to construct a setting where we'd expect planning-based and planning-free methods should make different predictions; I didn't see any experiment like that.

**Minor comments**:
* Run a spelling + grammar checker.
* "they serve as world models for goal-conditioned planning" -- Why can sequence models not serve as world models for more general RL problems? (Maybe cut "for goal conditioned planning")
* L65 -- I don't think "goal-conditioned predictive coding" has been defined yet
* When citing multiple papers, use "\citep{smith,adams}" rather than "\citep{smith}\citep{adams}". This removes the extra square brackets between the numbered citations.
* Don't use "etc" in technical writing.
* L94 Avoid using the passive voice.
* L111 Consider citing prior work that proposed very similar methods (albeit under different names). E.g., https://arxiv.org/abs/1906.05838, https://arxiv.org/abs/1912.06088
* L114 Avoid starting sentences with notation or numbered citations. Starting sentences with Author-year citations is fine, because they will read like a normal sentence: "Smith showed that ...".
* Alg 1 L3 -- Replace "{" with "(" to indicate that the trajectory is a sequence, rather than a set.
* Fig 1 -- Label the bottleneck.
* L182 "which" -> "Which"
* Fig 3 caption "left to right" duplicated.
* Table 6 caption, "best overall performance" --> "best average performance". I think it's a bit misleading to say "overall" because, on both tasks, there is a baseline that beats the proposed method.
* In all tables, bold the best numbers (and those that fall within the error bars).
* Citation [18] is a duplicate of [17]

**Questions:**

1. L133 "trajectory representations" -- Why are there multiple representations?
2. L136 -- What does it mean for B to be None?
3. Eq. 2 -- Is the policy conditioned on the entire trajectory, or just the current observation (as indicated in L14 in Alg 1, Figure 2)?
4. Alg 1, L3 -- Is $g$ the commanded goal or the goal that was actually reached in the future?
5. What is the motivation for the slot tokens?
6. The caption for Fig 1 says that the decoder will take in only the encoded slot tokens. Doesn't it also take in the encoded (and masked) history states?
7. In the caption for Figure 2, what are "latent future representations"?
8. L170, "remove the actions from pre-training inputs" -- When training the policy in Stage 2, does the encoder take the (past) actions are input?
9. In Table 5, the numbers for the _reproduced_ RvS-G method are a fair bit better than the original RvS paper on all but one task. What are the differences between the implementations that might explain this gain?
10. In a number of places, the paper refers to future representations and future information (e.g., L278). I'm confused what this is referring to. Is this the goal at the end of the trajectory, or something ese?

**Limitations:**

The limitations section does mention some limitations. I think there might be some other limitations that are a bit more important (e.g., the proposed method is much more complex than methods that skip Stage 1); the paper doesn't offer any theoretical explanation for _why_ the representations should help (If the tasks are Markovian, why should representations from a non-Markovian model help?).

---

> ### Author Rebuttal · Authors · 2023-08-10
>
> Thank you for your constructive feedback. We address your concerns in detail below and will update our paper accordingly:
> >***Some of the confusion is from the different masking strategies, which I didn't see explicitly defined anywhere.***
>
> We discussed the masking strategies in section 3.3 and also briefly introduced the definitions in Table 1. We will update section 3 to offer an elucidated clarification of these strategies. Specifically, in “AE-H” and “MAE-H”, we perform (masked) autoencoding solely on history sequence. “MAE-F”, “MAE-RC” and “MAE-ALL” requires TrajNet to reconstruct the entire trajectory including both history and future sequences. The differences between them are illustrated in the table below.
>
> |     	| MAE-F    	| MAE-RC      	| MAE-ALL     	|
> |---------|--------------|-----------------|-----------------|
> | History | Unmasked 	| Randomly masked | Randomly masked |
> | Future  | Fully masked | Fully masked	| Randomly masked |
> >***Unsure about the claims about implicit planning.***
>
> In section 4.3,  when we remove the goal-conditioning from the TrajNet, the bottleneck tries to predict the “goal-less” future and the planning can hardly be correctly performed (but we still keep the goal in PolicyNet). We can consider this "No condition" setting in Table 2 as a planning-free method. By comparing “No condition” and “Cond. on goal”, we observe that goal-conditioned bottleneck representations can significantly help policy learning. It suggests that this bottleneck encodes the desired future trajectory information according to the specified goal by performing the planning implicitly. In section 4.4 we further compare the bottleneck representation with the plan expressed as a sequence of explicit future states. Employing bottleneck representation yields a distinct performance improvement, which demonstrates that the planned future information can be well stored in a compact representation, and consolidates the role of the bottleneck as an “implicit planner”. In Figure A2, A3 and A4 of the supplementary, we also visualize the future states that are encoded in the bottleneck under different goal conditioning settings and the qualitative observations also align with above results.
> >***What does it mean for B to be None?***
>
> In our framework, the two stages can be connected via compact representations (B), or transferred architecture and weights. We consider B in the second case to be None.
> >***Is the policy conditioned on the entire trajectory or just the current observation?***
>
> In our design, PolicyNet has the observed trajectory but without architecture assumptions. An MLP policy conditions on the current observation by default, while a Transformer sequence model policy can consider the entire trajectory. In GCPC, the MLP policy is conditioned on the goal, current observation, and bottleneck, which either summarizes history, predicts future, or handles both tasks.
> >***Is the commanded goal or the goal that was actually reached in the future?***
>
> We follow RvS[10] to sample a reachable future state (or the return-to-go achieved in some number of future steps) as the goal. We will clarify this in both section 3.1 and Algorithm 1, and add the detail of the goal sampling process in the supplementary material.
> >***When training the policy in Stage 2, does the encoder take the (past) actions are input?***
>
> Specifically, we remove the actions from the TrajNet inputs during bottleneck computation. Since the TrajNet is frozen during the second stage, we keep the same input formulation as in the first stage. We will clarify this in the final version.
> >***The proposed method is much more complex than methods that skip Stage 1***
>
> We agree that our two-staged pipeline could be more computationally intensive than those policies that do not leverage sequence modeling. As shown by our experiments, properly utilizing sequence modeling would bring substantial improvements over multiple tasks. Therefore, we believe our approach is desirable for scenarios that can afford more computational costs. We also welcome further discussions and suggestions on the limitations of our approach.
> >***If the tasks are Markovian, why should representations from a non-Markovian model help?***
>
> Antmaze as a non-Markovian task due to its partial observability. In this environment, the agent relies on past experiences to infer its surroundings and formulate optimal decisions for subsequent steps. Both Kitchen and Gym Locomotion are Markovian tasks as they provide the global state information to the agent. In our design, GCPC can generalize to both scenarios, when the context length shrinks to 1, GCPC can be considered as a Markovian model. To understand how history would affect Makovian tasks, we perform ablation experiments on Kitchen by varying the context length within the “AE-H” objective (results are shown below). We observe that having history context can also benefit Kitchen tasks, which is similar to the case in DT[6]. One hypothesis is that during training, history conditioning helps the model identify the policy used to generate the corresponding actions, among a distribution of policies, thus improving the training dynamics. Moreover, sequence modeling can also be used for predicting the future. The predicted future information, such as waypoints/subgoals, has been shown to be helpful by previous works in both Makovian[a] and non-Markovian[b] tasks
>
> | AE-H  (Two-stage w/o actions) | 1          	| 5        	| 15        	|
> |-------------------------------|----------------|--------------|---------------|
> | Kitchen-Mixed             	| 62.0 (± 18.15) | 72.5 (± 3.0) | 72.5 (± 15.8) |
>
> [a] Jayaraman, Dinesh, et al. "Time-agnostic prediction: Predicting predictable video frames." arXiv preprint arXiv:1808.07784 (2018).
>
> [b] Chane-Sane, Elliot, Cordelia Schmid, and Ivan Laptev. "Goal-conditioned reinforcement learning with imagined subgoals." International Conference on Machine Learning. PMLR, 2021.

---

> > ### Comment · Reviewer_EZxU · 2023-08-10
> > **Reviewer response**
> >
> > Dear authors,
> >
> > Thanks for the thought and effort put into the rebuttal! The clarifications and questions largely made sense (unless otherwise noted below); please make sure to revise the paper for the clarifications. My main concern now is about the claims about implicit planning. The other points below are much more minor.
> >
> > > Implicit planning
> >
> > I’m not sure that the experiments in Sec 4.3 necessarily show that the proposed method is performing implicit planning. I agree that Table 2 shows that including the goal as an input is important, but I’m not sure that means that internally the method is performing planning.
> >
> > I’m also note sure that the experiment in Sec. 4.4 demonstrates implicit planning either. I agree it shows that conditioning the policy on the learned representation is better than conditioning the policy on future states predicted from that representation. To me, it seems hard to disentangle whether the representation is encoding information about the past (indeed, it only receives as input the past states) or the future. One could argue that sufficient statistics of the past are indistinguishable from the future; I would take that argument to mean that it’d be very challenging to definitively show that _any_ method is doing planning (vs history compression).
> >
> > > Limitations
> >
> > My concern here was not about whether the complexity was warranted, but rather about the limitations section seeming a bit shallow.
> >
> > > Antmaze is non-Markovian
> >
> > Can the authors elaborate on this point?
> >
> > > history conditioning helps the model identify the policy used to generate the corresponding actions
> >
> > This seems plausible to me, but I’m not sure why it would yield a better policy — by definition, in a Markovian environment the optimal action for the current state doesn’t depend on how actions were sampled in the past.

---

> > > ### Author Response · Authors · 2023-08-11
> > > **Response to Reviewer EZxU's response**
> > >
> > > Thank you for your prompt response! We'd like to further address your remaining concerns below:
> > > >***Implicit planning***
> > >
> > > We understand that the definition of “planning” could vary in different contexts and subtle ways. In our approach, the way we search for an optimal path may not be explicitly optimizing a certain metric, such as the distance or cumulative rewards. Regardless, our approach is capable of creating a "plan" that the agent will use to reach the "goal". In section 4.3, we want to show that simply predicting an arbitrary possible future (“No Condition” setting) would not be enough to help the agent solve the task. By specifying the goal in TrajNet, our model can generate a valid “plan” – a (sub-)path towards the goal (Figure A2, A3, and A4 in our supplementary), and significantly help decision making (quantitative results in Section 4.2, 4.3 and 4.4). We believe this process of generating a “plan”, which is helpful for achieving the goal, can generally be considered as “planning”.
> > >
> > > Regarding the argument about the entanglement of history and the future, we agree that the current way of representation learning would inevitably encode history information. However, in section 4.2 we’ve shown that merely encoding history (e.g. AE-H, MAE-H) cannot help solve this long-horizon task, while modeling the goal-conditioned future (e.g. MAE-F and MAE-RC) yields a substantial performance gain. We attribute this to the implicit planning performed by our model, which provides a latent plan (the bottleneck) that improves policy performance.
> > > >***Limitations***
> > >
> > > We agree that the limitation section can be further improved. Here we provide several failure cases/downsides of our approach in detail: (1) Our approach can be sensitive to the failure mode in the datasets, as GCPC predicts the future by fitting these offline collected trajectories, which includes some harmful behaviors. For example, in Antmaze, we note that some trajectories record actions that lead to the ant flipping over or constantly colliding with the wall. During the evaluation, we also observed our policy trained on these datasets could generate similar behaviors and even cause task failures. One possible solution to this issue is to introduce reward bootstrapping, which may assign low values to these behaviors and avoid them while performing the task. (2) Our approach cannot handle stochastic environments, since we always perform the greedy search with maximum likelihood estimation and could predict overly optimistic behaviors, which is undesired in stochastic environments.
> > > >***Elaborate why Antmaze is non-Markovian***
> > >
> > > Antmaze is a partially observable environment where the ant is only aware of its absolute (x,y) position in the maze and its proprioception (e.g. joint angle, joint velocity). The obstacle (e.g. the wall) location remains unknown to the ant. To perform the task with optimal decisions, the ant will need to understand whether a certain direction is blocked or not. By analyzing history, the ant can know whether it is stuck at or collided with an obstacle (e.g. the positions will keep the same before and after the collision), try to change direction by adjusting body orientations, and make better decisions for the following steps. So in this task, history states provide the necessary insights for optimal decisions, which makes this environment non-Markovian.
> > > >***Why does understanding history helps yield a better policy***
> > >
> > > We agree that ideally, past experiences wouldn’t have any impact on policy performance in a Markovian environment. But in practice, we indeed observed some benefits by applying the sequence modeling on the history trajectories, as in DT[6] they observed longer context length leads to improvements. The hypothesis we provide in the response mainly considers the history information serves as a form of regularization that helps regularize the complex policy distribution learned by the sequence model, which may yield better empirical performance.

---

> > > > ### Comment · Reviewer_EZxU · 2023-08-11
> > > > **Reviewer response**
> > > >
> > > > Dear authors,
> > > >
> > > > Thanks again for the detailed response! I know that the review period can be exhausting, so I really appreciate the time and effort put into these responses.
> > > >
> > > > I am still stuck up on the point about implicit planning. I think part of the challenge is that, exactly as pointed out in the rebuttal, "planning" can mean different things in different contexts. In some contexts, it means the ability to combine pieces of different trajectories to solve unseen tasks. In other contexts it means the ability to combine pieces of different trajectories to find new/better ways of solving previously seen tasks. In yet other contexts it refers to an explicit mechanism to infer what intermediate states should be visited when attempting to reach high-reward states. I think it would help to have a concrete definition of "planning." This definition should include things that obvious do planning (e.g., MPC, RRT, some model-based RL methods) and not things that don't clearly do planning (e.g., simple policy gradient and imitation learning methods).
> > > >
> > > > Part of the reason why I think the bar should be very high for demonstrating this is that, fundamentally, the paper is making a rather profound claim: that planning (a very sophisticated sort of behavior) can emerge from applying a simple objective (conditional imitation) with a particular architecture on a particular representation. It is also a very hard claim to prove. Here's one paper that gets at the sort of experiments I'd like to see to be convinced that a method is actually doing planning: "On the Utility of Learning about Humans for Human-AI Coordination"
> > > >
> > > > It's worth noting that this concern has very little to do with the actual method presented in the paper, only with the presentation of that method. If all claims to "planning" were cut from the paper, I would likely vote for accepting the paper.
> > > >
> > > > Minor notes:
> > > > * The revised limitations section looks great!
> > > > * Antmaze is non-Markovian -- This argument doesn't make sense to me. The position of obstacles is fixed, so the (x, y) position is sufficient for inferring the relative location of obstacles

---

> > > > > ### Author Response · Authors · 2023-08-12
> > > > > **Response to Reviewer EZxU's response**
> > > > >
> > > > > Thank you for elaborating on your concerns about implicit planning. We agree that the definition of “planning” needs to be well clarified and the profound claim of “implicit planning” remains to be supported by further experiments. Instead of claiming our method performs “obvious planning behaviors”, we would like to address your concerns by demonstrating the capability of our method to generate “implicit plans” that effectively help long-horizon decision making.
> > > > >
> > > > > We define an "implicit plan" as a goal-conditioned latent representation that can be decoded into a sequence of states (and actions) to reach the goal. The closest example to our definition is the "latent plan" term coined by the authors of "Learning Latent Plans from Play"[a], but left undefined. In their paper, the authors leverage a variational autoencoder to encode state and action sequences into a latent representation, which is then decoded to generate the next actions to take toward a goal. Both "implicit plan" and "latent plan" refer to a compact latent representation of a goal-reaching behavior, which can be expressed as a sequence of future state/action transitions heading toward the goal. However, in our two-stage design of TrajNet and PolicyNet, we do not assume a priori that the latent representation from TrajNet serves as an implicit plan, and it is up to PolicyNet how to leverage TrajNet's latent representation.
> > > > >
> > > > > Following this definition, the latent representations learned by MAE-F and MAE-RC (both with goal-conditioning) qualify to be candidates of "implicit plan": The latent representations encoded by TrajNet can be decoded to future states (or a state-action sequence) that move to the specified goal. Table 1 demonstrates that on Antmaze-Large, the pretraining objectives capable of learning "implicit plans" indeed outperform the other candidates. As discussed earlier in our rebuttal, Table 2 aims to study the impact of breaking "implicit plans" by removing the goal conditioning from TrajNet, and Table 3 aims to compare two ways of leveraging the implicit plans, either implicitly as a latent representation, or explicitly as the decoded future state (state-action) sequence.
> > > > >
> > > > > We fully agree with the reviewer that our current wording should be clarified to avoid confusion. We also agree with the reviewer that our experiments are not sufficient to demonstrate that the mechanisms of composing an implicit plan are the same as (explicit) planning, which we define as the process of leveraging a world-model to search for the actions that reach the states with high rewards or Q-values. As opposed to "planning can emerge from applying a simple objective", what we have demonstrated is that the pretraining objectives that learn "implicit plans" (per the definition above) are the most effective sequence modeling strategy. We will clarify this in the introduction and experiment section 4.3 in our revision. In addition, we welcome your suggestion on an alternative naming for "implicit planner" to better avoid confusion.
> > > > >
> > > > > [a] Lynch, Corey, et al. "Learning latent plans from play." Conference on robot learning. PMLR, 2020.

---

> > > > > > ### Comment · Reviewer_EZxU · 2023-08-13
> > > > > > **Reviewer response**
> > > > > >
> > > > > > Dear authors,
> > > > > >
> > > > > > Thanks for continuing the discussion!
> > > > > >
> > > > > > I appreciate the formal definition. I think it's a step in the right direction, but think that there's still some ambiguity about what "decoding" means. As an example, what if we defined the tuple $(s, g)$ as the implicit plan, and defined the decoding strategy as some explicit planning. I think this accords with the formal definition, but it implies that the tuple $(s, g)$ is an implicit plan. This is true in a literal sense -- it is a sufficient statistic for the conditional distribution p(intermediate states | start, goal). But it seems to violate our intuition about what planning really means -- the identity map does not do planning. So, I think that a formal definition would likely need some sort of restrictions on what decoding strategies are allows.
> > > > > >
> > > > > > Here's one question to get started: do both the encoder and decoder reason about _time_?
> > > > > >
> > > > > > I agree that prior work has also used the work "planning" somewhat vague ways.

---

> > > > > > > ### Author Response · Authors · 2023-08-16
> > > > > > > **Authors' response**
> > > > > > >
> > > > > > > Dear Reviewer EZxU,
> > > > > > >
> > > > > > > Thank you and sorry for our delayed response!
> > > > > > >
> > > > > > > This is a great question, and we really appreciate it! $(s, g)$ can indeed be considered as a special case of an _implicit plan_ per our initial definition, and we agree that such corner cases make the definition less informative.
> > > > > > >
> > > > > > > Please allow us to refine the definition of an implicit plan as follows: An implicit plan is a _compact_ latent representation that contains _sufficient information_ to construct a decoder (e.g. a probabilistic generative model) that can generate goal-reaching state (and action) sequences with high probabilities, without relying on "side information" from the environment. Under this revised definition, simply using $(s, g)$ as an implicit plan would not work, as all trajectories that start from $s$ and $g$ would be deemed equally likely (or unlikely) by the decoder which does not have access to a _world model_ of an environment. To answer your question, we believe the encoder needs to have the notion of _time_, and needs to reason about the future (and the past), while the decoder may not. We conjectured that the goal-conditioned predictive coding objective may serve as an implicit plan as it aims to learn a compact representation that can be decoded into **a** future sequence towards the goal, but we agree additional experiments would be needed to validate our conjecture (e.g. if the latent representation is sufficient, as opposed to capturing only _surface statistics_).
> > > > > > >
> > > > > > > We also agree with the reviewer that having a rigorous definition of implicit planning, though very useful, is not directly connected to the main contributions of our paper. We introduced the term with the hope to provide an intuitive explanation of our conjectures on the effectiveness of GCPC. We are happy to avoid the confusion by (1) revising the use of "implicit planner" term from the title and in the main paper; (2) moving the related discussion into appendix where we have more space, and incorporating our discussions above.
> > > > > > >
> > > > > > > Again, we really appreciate your thoughtful comments and constructive feedback :)
> > > > > > >
> > > > > > > Best,
> > > > > > >
> > > > > > > The Authors

---

> > > > > > > > ### Comment · Reviewer_EZxU · 2023-08-16
> > > > > > > > **Reviewer resposne**
> > > > > > > >
> > > > > > > > Dear authors,
> > > > > > > >
> > > > > > > > Thanks for continuing the discussion! If the paper were revised to remove all instances of "plan/planner/planning" from the main text, I would vote for accepting the paper. E.g., the title could be something like "Goal Conditioned Predictive Coding for Offline RL". Maybe one exception would be a sentence in the method or experiments section saying "We analyze whether GCPC performs planning in Appendix X."

---

> > > > > > > > > ### Author Response · Authors · 2023-08-16
> > > > > > > > > **Authors' response**
> > > > > > > > >
> > > > > > > > > Dear reviewer EZxU,
> > > > > > > > >
> > > > > > > > > Yes, we will make your suggested changes on "plan/planner/planning" in the final version. Thank you so much for the super valuable discussions on this!
> > > > > > > > >
> > > > > > > > >
> > > > > > > > > Best,
> > > > > > > > >
> > > > > > > > > The Authors

---

> > > > > > > > > > ### Author Response · Authors · 2023-08-18
> > > > > > > > > > **Authors' response**
> > > > > > > > > >
> > > > > > > > > > Dear reviewer EZxU,
> > > > > > > > > >
> > > > > > > > > > Could you please let us know if our responses above are able to address your concerns? Thank you!
> > > > > > > > > >
> > > > > > > > > > Best,
> > > > > > > > > >
> > > > > > > > > > The Authors

---

> > > > > > > > > > > ### Comment · Reviewer_EZxU · 2023-08-18
> > > > > > > > > > > **Reviewer response**
> > > > > > > > > > >
> > > > > > > > > > > Dear authors,
> > > > > > > > > > >
> > > > > > > > > > > I don't have additional concerns -- the rebuttals have been fantastic in this regard!
> > > > > > > > > > >
> > > > > > > > > > > I will discuss with the other reviewers their concerns, and their predictions about whether the "planning" changes will actually be made to a camera-ready version (if it were possible to see a version of the paper that had cut "planning", that would increase my confidence on this point).

---

> > > > > > > > > > > > ### Author Response · Authors · 2023-08-18
> > > > > > > > > > > > **Authors response**
> > > > > > > > > > > >
> > > > > > > > > > > > Dear Reviewer EZxU,
> > > > > > > > > > > >
> > > > > > > > > > > > Thank you!
> > > > > > > > > > > >
> > > > > > > > > > > > As we cannot provide a revision as part of the rebuttal, please allow us to present our plan to address the concerns on "implicit planning" in the revision as follows:
> > > > > > > > > > > >
> > > > > > > > > > > > - Title: "Goal-Conditioned Predictive Coding for Offline Reinforcement Learning"
> > > > > > > > > > > > - line 13 and line 68: remove _serves as an “implicit planner”_
> > > > > > > > > > > > - line 77: remove _and perform implicit planning_
> > > > > > > > > > > > - line 159: revised as _Our intuition is ... and may perform implicit planning (see Appendix for detailed discussion)._
> > > > > > > > > > > > - line 183: remove _Is there any evidence ... implicit planning_
> > > > > > > > > > > > - line 231: change the subsection title as _impact of goal conditioning for trajectory representation learning_, and modify line 232-234, and caption of table 2, accordingly.
> > > > > > > > > > > > - line 250: remove _this result can also be considered ..._, and modify the caption of table 3 accordingly
> > > > > > > > > > > > - line 280: remove _we show that the pretrained bottlenecks have the capabilities to perform implicit planning_
> > > > > > > > > > > >
> > > > > > > > > > > > Again, please allow us to thank you and all the reviewers for sharing your detailed and constructive feedback, which truly helps us improve our paper. We have been incorporating the feedback into our revised manuscript, and we look forward to presenting our revised manuscript in the camera ready version!
> > > > > > > > > > > >
> > > > > > > > > > > > Best,
> > > > > > > > > > > >
> > > > > > > > > > > > The Authors

---

### Official Review · Reviewer_eFoi · 2023-07-03

**Soundness:** 2 fair
**Presentation:** 2 fair
**Contribution:** 3 good
**Rating:** 5
**Confidence:** 4

**Summary:**

This work introduces Goal-Conditioned Predictive Coding (GCPC) – a goal conditioned policy learning method that incorporates trajectory representations learned by a bi-direction transformer. The authors introduce a two-phase framework for goal-conditioned learning, consisting of a trajectory encoding phase, and a policy learning phase. GCPC uses a bidirectional transformer with different masking schemes as the trajectory encoder, then feeds the representation from fixed bottleneck tokens to a bidirectional decoder to predict the trajectory. Then, the policy is learned using the current state, goal, and trajectory representation. The author’s primarily evaluate their approach by taking the table from RvS (Emmons et al.) and adding their method, GCPC. They also provide a number of ablation studies over different masking strategies etc.


**Strengths:**

This paper presents an original two phase method for learning policies from offline data using supervised methods. The idea of decoupling trajectory representation via a bottle-necked bidirectional transformer is new, though bottlenecked representations have been studied before with other architectures, like VAEs. It's good to see this considered as well.

The author’s ablation studies on their method are extensive – they compare different masking schemes, using and not using actions, the importance of goal representations, and window size. The quantity of datasets considered is also high.

I also found the paper to be well organized, though with a few gaps in writing.


**Weaknesses:**

**Experiments**

My biggest concerns with this work are regarding the experimental comparisons in Section 4.6 – the primary results tables.

Specifically, *the baseline results are copied from other papers*  (RvS, SPOT). This is not stated in the main text or in the table caption (it should be, please add it there!). While I usually think it's better when authors run their own experiments, copy-pasted tables are now prevalent in the offline RL community.  For this work, however, I believe there is more that should be done.

First, it seems like the authors are comparing goal conditioned algorithms to unconditional algorithms. For example, in Table 5 on Ant Maze BC, DT, TD3+BC, CQL, and IQL are all not goal conditioned (the paper from which these numbers are pulled does not train goal conditioned policies). On the other hand, RvS-G and GCPC, the two best performing methods on the table are goal conditioned! This feels problematic as in Table 2 its explicitly shown that without goal conditioning on Ant Maze GCPC drops from scores of above 75 to scores below 20 (50 point difference!). The same can be said for Table 6 with the kitchen environments, which might benefit from goal-conditioning. The effect of this is that the main experimental results feel like an apples to oranges comparison.

Second, the baselines chosen for the goal-conditioned experiments seem inappropriate. For goal-conditioned environments, the authors should be testing against goal-conditioned algorithms: Goal-Conditioned IQL with hindsight relabeling, decision transformer with goal-conditioning, GSCL [1], WGCSL [2], DWSL [3]. The best performing method on many environments in the RvS paper is GCSL as it has some degree of policy improvement via relabeling, but they don’t run it on most environments!

Finally, there is no notion of how much more computationally expensive GCPC is than baselines. GCPC has two stages, and one of those stages involves training a large bidirectional transformer. Many baselines, on the other hand, train only small MLPs.

To fix these issues, I think the authors should:
1. Make it clear they take experiment numbers from prior works.
2. Make it clear when experiments are goal conditioned or not
3. For experiments that are goal conditioned, run baselines that are also goal-conditioned, ie GCSL, GC-IQL, Goal Conditioned DT, DWSL, testing hindsight relabeling as appropriate.
4. Add a (short) discussion of compute intensity.

**Two-Stage Framework**

The authors attempt to cast all “transformer for offline RL” methods into a general two stage framework. While I sort of see how this works, I don’t generally buy this as a cohesive unified framework. Most prior approaches all use the same transformer network for both trajectory representation and the policy (DT, MTM, TT). Casting other methods into this framework feels a bit unnatural, as no approach except for GCPC actually uses both a separate TrajNet and PolicyNet. I think that’s totally fine! But, I think the paper would be better explained by just stating that most other approaches use the same network for sequence representation and the policy, and GCPC separates them. Section 3.2 just doesn’t feel like the unified framework its supposed to be.

**Writing**

I found aspects of the writing confusing, and I believed this can be improved in the next version:
1. The definitions of each masking scheme aren’t given until Table 1 in the results section. Given that you discuss these earlier in the paper, it feels necessary to define them in Section 3.
2. Updating Figure 1 & 2 to make it more clear that only the bottleneck states are passed between the encoder and decoder. Also, what are the other inputs to decoder? Does it have the bottle neck tokens plus all the mask tokens for each state and action?
3. Generally many important details – experiment numbers are taken from other papers, what is bn in Table 2, etc. – are unanswered or left in the appendix.
4. The authors use (s, a, g) tuples in their writing, but state that sometimes g is just the reward. In the experiments section, it's really unclear when g is a state goal or the reward.
5. I would use something other than Q for slot tokens.

[1] Ghosh, Dibya, et al. "Learning to Reach Goals via Iterated Supervised Learning." International Conference on Learning Representations. 2020.

[2] Yang, Rui, et al. "Rethinking Goal-Conditioned Supervised Learning and Its Connection to Offline RL." International Conference on Learning Representations. 2021.

[3] Hejna, Joey, Jensen Gao, and Dorsa Sadigh. "Distance Weighted Supervised Learning for Offline Interaction Data."  ICML 2023.


**Questions:**

What does bn mean in table 2? Is it batch norm? If so, it seems a bit unfair to use batchnorm for your method, while none of the baselines use it.

For which datasets / environments is GCPC goal conditioned? When is it using reward? This should be in the paper.


**Limitations:**

The authors could do a better job providing examples of failure cases or downsides of GCPC, if they exist. The limitations section as is mostly contains directions for future work.

---

> ### Author Rebuttal · Authors · 2023-08-10
>
> Thank you for your constructive feedback. We address your concerns in detail below and will update our paper accordingly:
> >***Make it clear when numbers are quoted from published results***
>
> We completely agree! We will incorporate the reviewer’s suggestion and explicitly explain if a baseline’s performance is quoted from their original papers.
> >***For which datasets/environments is GCPC goal conditioned? When is it using reward?***
>
> For the conditioning variable, we followed the setup in RvS paper: In Antmaze and Kitchen, we used the target state as goal; In Gym locomotion, we used rewards (return-to-go). We will clarify this in the final version.
> >***It seems like the authors are comparing goal-conditioned algorithms to unconditional algorithms.***
>
> We acknowledge that additional details, such as if an algorithm is goal-conditioned, should be highlighted, and we will incorporate the change in the final version. We follow prior work (TT [17], DD [1], RvS [10]) to present a variety of methods, including supervised learning methods (e.g. conditional BC, sequence modeling, etc.) and valued-based methods, for a comprehensive comparison. In the rebuttal PDF Table R4, we evaluate two suggested goal-conditioned methods DWSL[b] and WGCSL[a]. Our proposed GCPC outperforms DWSL and WGCSL by large margins. Please find our next response for more details.
>
> In Table 2 of our original paper, the comparisons are not intended to demonstrate the superiority of goal-conditioning compared to alternative learning paradigms. Rather, it is an ablation study to demonstrate the importance of goal-conditioning for GCPC’s trajectory representation learning. We alter the inputs to TrajNet to include or exclude goal conditioning, and pass the resulting bottleneck representation to PolicyNet. In either scenario, PolicyNet takes the current state, the bottleneck and the goal as its inputs, and is thus always goal-conditioned. We will also clarify this detail in our revised paper.
> >***The authors should be testing against more goal-conditioned algorithms.***
>
> We thank the reviewers for suggesting highly relevant works and agree that comparisons with goal-conditioned methods are needed for both Antmaze and Kitchen. Here we’ve provided two additional supervised goal-conditioned baselines – DWSL[b] and WGCSL[a]. For both algorithms, we use the implementation provided by DWSL’s authors, set the hindsight relabeling ratio to 1.0 to recover the experimental setting adopted by both RvS-G and GCPC, set training hyperparameters as recommended in the paper appendix[d], and re-run the baseline experiments under the same evaluation setup as ours. Please find the results in Table R4 of our attached PDF file. We observed that the performances of both goal-conditioned baselines are generally lower than the numbers reported by the DWSL paper. By comparing training curves and result tables (both from the DWSL paper), we speculate that their results correspond to the best mean performance during the whole training process, whereas our setup focuses on measuring the performance achieved near the end of the training. Among all supervised goal-conditioned baselines, GCPC significantly outperforms the compared ones on average performance and has a substantial improvement over Antmaze Large and Kitchen environments.
> >***Add a (short) discussion of computing intensity.***
>
> We will add a short discussion of computing intensity compared to planning-based methods (e.g. TT[17] and DD[1]) in the revised version, mainly focusing on the following three aspects:
>
> 1) As a sequence modeling approach, TT encodes (state, action, reward) tuples as $(|S| + |A| + 1)$ tokens, where $|S$| and $|A|$ refer to the dimensions of state and action spaces number. GCPC needs only $1$ state token per timestep. Assuming that hidden dimensions of each token are the same for both methods, the complexity of TT is $(|S| + |A| + 1)^2$ times that of GCPC, which will sharply increase when scaling to high-dimensional state/action spaces.
>
> 2) Both DD and TT require either iterative sampling or step-by-step prediction to obtain a sequence of future states during evaluation. By default, for each evaluation step, DD takes $200$ diffusion steps to obtain a denoised state sequence, TT needs to predict 15 future timesteps ($15 * (|S| + |A| + 1)$ tokens) autoregressively. GCPC predicts future information within $1$ model forward step, which yields a significant efficiency gain.
>
> 3) In practice, we recorded the wall time of performing evaluation on Antmaze Large using GCPC and DD. We observed DD takes nearly 1 hour for 1000 evaluation steps while GCPC takes no more than 30 seconds (a **120x** speedup).
>
> >***Updating Figure 1 & 2 to make it more clear. What are the other inputs to decoder?***
>
> In the actual implementation, the decoder will take bottleneck tokens and a sequence of placeholder tokens for the states. We will update the corresponding figures in the final version.
> >***What does bn mean in table 2?***
>
> "bn" refers to the bottleneck. Since in both setting PolicyNet takes bottleneck as input, this notation is redundant and we will remove it from Table 2 for clarity.
>
> >***Providing examples of failure cases or downsides of GCPC, if they exist.***
>
>  During evaluation, we observe our model would generate failure behaviors that appear in the dataset (e.g. the ant would flip over caused by bad actions) and is sensitive to the out-of-distribution states, which lead to suboptimal actions. In addition, our current approach could fail when stochasticity exists in the environment dynamics since GCPC always performs greedy planning with maximum likelihood estimation.
>
>
> [a] Yang, Rui, et al. "Rethinking Goal-Conditioned Supervised Learning and Its Connection to Offline RL." International Conference on Learning Representations. 2021.
>
> [b] Hejna, Joey, Jensen Gao, and Dorsa Sadigh. "Distance Weighted Supervised Learning for Offline Interaction Data." arXiv preprint arXiv:2304.13774 (2023).

---

> > ### Comment · Reviewer_eFoi · 2023-08-10
> > **Rebuttal Acknowledged**
> >
> > I would like to thank the authors for their efforts in updating the manuscript, especially in adding additional experiments. Consequently, I have decided to raise my score.
> >
> > Thank you for also pointing out that the DWSL authors use a different evaluation procedure -- it does seem like they take the max( avg (seeds)) -- loosely justified because DWSL is supervised and can overfit.
> >
> > I am a bit surprised DWSL does worse than RvS-G, but I guess this is probably either because value learning hurts or a difference in the codebases. I would request that in the appendix the authors include a list of which codebases they used to get which results.
> >
> > I am also suprised based on Figure R1 in the authors response that they did not use GC-IQL as a baseline for Antmaze -- DWSL cites Antmaze as a "failure case" for their method, and GC-IQL seems to perform better in the learning curve and is based on a more standard algorithm.
> >
> > It would be nice if the above points could be addressed in the final paper.

---

> > > ### Author Response · Authors · 2023-08-13
> > > **Response to Reviewer eFoi's response**
> > >
> > > Thank you for your prompt reply and suggestion to compare with GCIQL!
> > >
> > > We have carefully examined the reproduced codebase and identified one inconsistency with the original WGCSL implementation: the negative sparse reward function needs to be explicitly specified for hindsight relabeling, otherwise the rewards will be all zeros. We also notice that although the DWSL’s config for Antmaze did not set the dropout rate, it uses the codebase’s default rate, while we set the rate to 0.1. After this discovery, we have confirmed that all the hyperparameters, including those not explicitly specified in the config, are consistent with the authors’ suggestions. We re-run additional baseline experiments and our updated results are as follows:
> > >
> > > |                | DWSL         | WGCSL         | GCIQL         | RvS-G         | GCPC         |
> > > |----------------|--------------|---------------|---------------|---------------|--------------|
> > > | Umaze          | 71.2 (± 4.2) | 90.8 (± 2.8)  | 91.6 (± 4.0)  | 69.0 (± 5.1)  | 70.0 (± 8.2) |
> > > | Umaze-Diverse  | 74.6 (± 2.8) | 55.6 (± 15.7) | 88.8 (± 2.2)  | 67.0 (± 6.3)  | 73.8 (± 7.3) |
> > > | Medium-Play    | 77.6 (± 3.0) | 63.2 (± 13.7) | 82.6 (± 5.4)  | 70.8 (± 5.2)  | 70.4 (± 4.1) |
> > > | Medium-Diverse | 74.8 (± 9.3) | 46.0 (± 12.6) | 76.2 (± 6.3)  | 74.6 (± 3.7)  | 70.8 (± 6.4) |
> > > | Large-Play     | 15.2 (± 7.7) | 0.6 (± 1.3)   | 40.0 (± 16.2) | 40.6 (± 11.2) | 79.2 (± 6.9) |
> > > | Large-Diverse  | 19.0 (± 2.8) | 2.4 (± 4.3)   | 29.8 (± 6.8)  | 25.2 (± 4.8)  | 77.2 (± 2.3) |
> > > | Average        | 55.4         | 43.1          | 68.2          | 57.9          | 73.6         |
> > >
> > > |                 | DWSL         | WGCSL         | GCIQL         | RvS-G         | GCPC         |
> > > |-----------------|--------------|---------------|---------------|---------------|--------------|
> > > | Kitchen-Mixed   | 61.9 (± 5.5) | 56.0 (± 5.8)  | 44.5 (± 13.2) | 40.0 (± 19.4) | 61.0 (± 6.0) |
> > > | Kitchen-Partial | 47.5 (± 6.7) | 49.5 (± 10.1) | 45.5 (± 11.8)  | 46.5 (± 15.9) | 65.0 (± 9.2) |
> > > | Average         | 54.7         | 53.0          | 45.0          | 43.3          | 63           |
> > >
> > > We observe the baseline results on Antmaze basically align with the trend of corresponding curves present in the DSWL paper, and we attribute the remaining performance differences to the discrepancy in the evaluation procedure. Additionally, we provide GC-IQL performance as suggested. We follow GCIQL’s suggested hyperparameters, and use our evaluation procedure. We observe GCIQL indeed performs competitively on Antmaze, especially on Umaze and Medium, which is consistent with the observations in the DWSL paper. GCPC is able to outperform DWSL, WGCSL, and GCIQL on Antmaze Large and Kitchen.

---

### Official Review · Reviewer_tnif · 2023-07-04

**Soundness:** 2 fair
**Presentation:** 2 fair
**Contribution:** 2 fair
**Rating:** 3
**Confidence:** 4

**Summary:**

The paper presents a perspective on RL via Supervised Learning (RvS) algorithms that identifies two steps—trajectory representation and policy learning—and proposes a specific algorithm for offline RL called Goal-Conditioned Predictive Coding (GCPC) based on this view. The trajectory encoder is modelled as Bidirectional Transformer (BT) that compresses trajectories into slot tokens, which are then passed to a goal-conditioned policy. Interestingly, actions are omitted from trajectories, which improves the performance. Ablation studies on the Antmaze environment on 3 seeds are conducted, including

1) masking objectives for BT pertaining (=> masking entire future and inputing (randomly masked) past + future works best)
2) impact of goal conditioning (=> conditioning helps a lot)
3) comparing latent vs explicit future prediction (=> as per Table 3, there is no significant difference (returns within error bounds))
4) effects of future window size (=> bigger better)

Comparisons against baselines on Antmaze-v2, Kitchen, and Gym Locomotion are provided. GCPC performs on the same level as Decision Diffuser (DD) (within error bounds) on Kitchen and Gym, and below IQL on Antmaze (Maze and Medium). Only on Antmaze Large GCPC outperforms all baselines.

**Strengths:**

- Originality: the paper presents a nice perspective on existing algorithms, that distinguishes between trajectory representation and policy learning steps. The exact proposed architecture is novel, but it is quite similar to alternatives.
- Quality: a number of ablation studies are provided and comparisons to many baselines. However, the ablations are done on 1 environment and on 3 seeds only, and many results are within error bounds, so it is hard to make conclusions. The comparisons to baselines do not show improvement over DD/IQL except on one task: Antmaze Large.
- Clarity: the paper is mostly clear, but improvements can be made. E.g., table captions can be more informative, many claims need to be toned down, e.g., "significant influence" of representation pertaining and "superior performance" claimed in the conclusion are not supported by the results.
- Significance: minor. The approach performs similar to IQL/CQL/DD. Practically, there is no significant advantage to it.

**Weaknesses:**

The idea seems nice but the evaluations are insufficient.
1) The ablations need more environments and seeds. It is not clear that ablations on Antmaze generalise to FrankaKitchen, for example.
2) Performance on Kitchen and Gym is not so different from the baselines. Only Medium dataset used on Gym. No DD on Antmaze.
3) See other comments from above about originality/quality/clarity/significance.

**Questions:**

- Antmaze seems like a quasi-stationary environment in the sense that the policy for locomotion does not depend on where in the maze the ant is. What is the meaning of compressing a trajectory in this case? It seems wasteful.

**Limitations:**

yes

---

> ### Author Rebuttal · Authors · 2023-08-10
>
> Thank you for your constructive feedback! We address your concerns in detail below and will update our paper accordingly:
>
> >***More seeds are needed for ablation experiments.***
>
> We reported ablation experiments on FrankaKitchen in the supplementary materials Table A3 and A4. In Table R1, R2, and R3 of the rebuttal PDF, we report ablation experiments of Antmaze and Kitchen with 5 seeds. We observe that our observations still hold after more seeds are used: On Antmaze Large, using goal-conditioned predictive coding (i.e. MAE-F and MAE-RC) and removing actions significantly outperform the alternative sequence modeling strategies. However, as we already observed in the supplementary, the trends on FrankaKitchen are less conclusive, as most performance gaps fall within error bounds. Moreover, the performance variances tend to become larger when more seeds are used. We attribute this to the much smaller trajectory dataset size of FrankaKitchen: Compared to Antmaze datasets (nearly 1000 trajectories and 1001 transitions per trajectory), Kitchen datasets are much smaller (nearly 500 trajectories, but only 227 transitions per trajectory on average), and provide one order of magnitude fewer training examples per epoch. We thus conjecture that the TrajNet is more prone to overfitting and may fail to encode trajectory information useful for planning and decision making.
>
> >***The performance is similar to IQL / CQL / DD. What are the practical advantages of the proposed method?***
>
> As shown in Tables 6 and 7 of the original submission and the evaluation on CALVIN[a] in our general response, GCPC outperforms both IQL[b] and CQL[c] consistently. Notably, on Antmaze Large, where most previous methods failed to solve this challenging task, GCPC achieves remarkable performance and surpasses the most performant baseline, IQL, by a large margin. While GCPC performs similarly to DD[1] on Kitchen and Gym, we note that DD is based on conditioned diffusion models, and our design is more efficient than DD during training and evaluation. On a single RTX3090 GPU, our approach achieves 3x speedup for training and **120x** speedup during evaluation on the AntMaze Large environment, when compared to the DD implementation provided by its authors. We believe the consistently competitive performance across diverse benchmarks, and the computational efficiency are the main practical advantages of our method.
>
> >***Originality of the proposed framework.***
>
> The primary goal of this paper is to understand how sequence modeling can be properly applied to decision making, and our framework is proposed for this purpose. To investigate the influence of sequence modeling on decision-making, we dissect the process into trajectory representation learning and policy learning. In the meantime, we aim to propose a general framework to compare recent works on sequence modeling for decision making, rather than a particular unique design. We extensively examined various sequence modeling strategies under the proposed two-stage framework, and discovered a powerful objective (GCPC) for trajectory representation learning. We demonstrate the importance of future planning in long-horizon tasks and validate the feasibility of acquiring an implicit planner – the bottleneck, with sequence modeling techniques. In addition, we further demonstrate the efficacy of the “implicit planner” by conducting a comparative analysis against the plans in the form of explicit future states. We believe our insights on the impact of sequence modeling for decision making, as well as the discovered GCPC objective with strong empirical performance, are both original and significant.
>
> >***Why only Medium dataset was used on Gym? How about DD performance on Antmaze?***
>
> We report medium dataset performance as it is a standard practice adopted by previous work, such as TT[17], and DD[1]. The rationale is that the medium dataset contains suboptimal trajectories which can be used to validate a method’s capability to learn better policies than the offline dataset.
>
> DD did not report on Antmaze and DD’s authors don't provide any hyperparameters for Antmaze either. As noted in our response on practical advantage above, we observe that training DD on Antmaze (with its default number of training steps) demands nearly 20 hours. It is thus impractical for us to perform a meaningful hyperparameter search and report its performance.
>
> >***Clarity can be improved, such as table captions. Many claims need to be toned down.***
>
> Thank you for the suggestions. We will elaborate table contents in captions and adjust some statements as appropriate based on the results in the revised paper
>
> >***Antmaze seems like a quasi-stationary environment in that the locomotion policy does not depend on where in the maze the ant is. What is the meaning of compressing a trajectory in this case? It seems wasteful.***
>
> In the Antmaze, the ants need to avoid getting stuck in the corners and choose the correct direction (i.e. by adjusting the body orientation) at each intersection point. Since Antmaze is a partially observable environment (the ant is only aware of whether a certain direction is blocked or not), the history information is needed for the ant to remember the obstacles. Our framework effectively acquires these helpful signals by compressing a trajectory into a compact representation. This compressed representation is subsequently leveraged for supporting policy learning, as it empowers the ant to gain the necessary insights for successful task completion.
>
> [a] Mees, et al. "Calvin: A benchmark for language-conditioned policy learning for long-horizon robot manipulation tasks." IEEE Robotics and Automation Letters 7.3 (2022)
>
> [b] Kostrikov et al. "Offline reinforcement learning with implicit q-learning." arXiv preprint arXiv:2110.06169 (2021).
>
> [c] Kumar, Aviral, et al. "Conservative q-learning for offline reinforcement learning." Advances in Neural Information Processing Systems 33 (2020)

---

> > ### Comment · Reviewer_tnif · 2023-08-10
> > **Rebuttal Acknowledged**
> >
> > Thank you for the replies and clarifications
> > 1.  More seeds: thanks for pointing out the tables in the appendix and for providing additional ablations in the rebuttal pdf. It is indeed somewhat concerning that "**the performance variances tend to become larger when more seeds are used**".
> > 2. "As shown in Tables 6 and 7 ... GCPC outperforms both IQL[b] and CQL[c] consistently" — I have to point out that this is factually not true. On Partial Kitchen, IQL (64.3) is well within the confidence interval of GCPC (65 ± 9.2); on Medium-Replay HalfCheetah, IQL (44.2) outperforms GCPC (41.1 ± 0.3); and there are more cases where they are within confidence bounds. So, sometimes GCPC better, but mostly it is comparable.
> > 3. I already acknowledged the notable performance of GCPC on Antmaze Large in my original review, so here I agree with the authors. The computational advantage over DD sounds very strong, it would definitely be important to mention it in the paper too.
> > 4. Originality of the framework: thank you for the additional arguments about the framework. I think the perspective of decoupling trajectory representation and policy learning provided by the paper is one of its big strength, and I also mentioned it in my original review already.
> > 5. Antmaze trajectory representation: I understand your response. What I was getting at is that perhaps Antmaze is not the best example to demonstrate your algorithm, because it seems the control policy for Antmaze can be completely decoupled between 2 levels: point navigation in a maze (for which you need to collect history) and limb control (which is the same irrespective of where the ant is in the map). So, in principle it is nice to see that one can just feed trajectories and learn something, but it seems a bit a made-up problem.

---

### Official Review · Reviewer_3wNA · 2023-07-06

**Soundness:** 4 excellent
**Presentation:** 4 excellent
**Contribution:** 4 excellent
**Rating:** 8
**Confidence:** 5

**Summary:**

The manuscript presents a novel framework for formulating decision making as a supervised learning problem on offline-collected trajectories. Contrary to previous methods, the authors propose to decompose the process into two stages: a) a trajectory representation learning stage (using sequential modeling techniques, like Transformers), and b) a policy learning stage where the trajectory representation learned in the previous stage is being used as an input to the policy. Overall, the proposed framework and specific instantiation (GCPC) is extensively evaluated and analyzed through experiments in 5 different simulation environments.

**Strengths:**

The main strengths of the manuscript are as follows:

- The paper is overall well written, the framework clearly articulated and the experiments well presented.
- I like the decoupling to two stages: it makes sense, makes things clear, and allows for better experimentation.
- Extensive experiments and analysis.


**Weaknesses:**

I have, however, a few concerns:

- Why did you use the success rate over 100 evaluation trajectories for the AntMaze and only 10 for the other scenarios?
- "We use the best evaluation result among the last five checkpoints as the performance for each seed." Is there a specific reason for using the best among the last five checkpoints and not just the last checkpoint? Did the learning have a tendency to diverge or oscillate?
- "For each experiment, we report the mean performance averaged over different seeds.". Using the median and quartiles/percentiles should on average be a better metric. See this wonderful paper by colleagues on how to properly choose the number of seeds and type of metric: https://arxiv.org/abs/1806.08295
- In section 4.3, did the authors remove both the goal and the bottleneck inputs? Or just the bottleneck ones? Or just the goal? There needs to be some clarification here in the text. If they removed both, I am not sure if this experiment has any importance.


Typos:

- Line 228: " Other than that, We also". W should be lowercase


**Questions:**

I do not have any important questions.

**Limitations:**

The authors adequately discuss the limitations of their approach.

---

> ### Author Rebuttal · Authors · 2023-08-10
>
> Thank you for your constructive feedback! We address your concerns in detail below and will update our paper accordingly:
>
> >***The numbers of trajectory rollouts***
>
> In our experiment, we regard IQL[a] as a robust value-based baseline, which has demonstrated its effectiveness across a diverse array of tasks and is widely compared by other works. We used the same number of evaluation rollouts as adopted by IQL, employing 100 rollouts for Antmaze and 10 rollouts for the other environments. We note that this setup is aligned with our empirical observations: With the same model checkpoint, the performance variance across different evaluation rollouts is often higher in Antmaze, than that of Kitchen and Locomotion.  At the earlier stage of our experiments, we applied more rollouts for Kitchen and Locomotion, and found their performances remained stable when using 10 or more rollouts.
>
> >***Why select from the last five checkpoints? Did the learning have a tendency to diverge or oscillate?***
>
> Yes, we indeed observed that performance may oscillate for the AntMaze environment. Similar behavior has been repeatedly observed in earlier works (e.g. WGCSL[b], DWSL[c]).  Please find the learning curves of GCPC and multiple previous goal-conditioned methods in Figure R1 of the attached rebuttal PDF file. The curves of previous goal-conditioned methods, including GCSL[d], WGCSL, GoFar[e], GCIQL, and DWSL, were reported by the DWSL paper based on 50 rollouts.  When overlaying these curves, we can clearly see that GCPC achieves the best performance among all approaches. Our evaluation strategy more faithfully captures the performance trends, which is particularly helpful for ablation studies. In Table R4 of the rebuttal PDF, we provide the results of DWSL and WGCSL using our evaluation metrics. Our GCPC outperforms both baselines by large margins.
>
> >***Using the median and quartiles/percentiles should on average be a better metric.***
>
> Thank you for the reference! We agree that better metrics, such as median and quartiles/percentiles, would make the evaluation results more fair and complete. We reported the average performance to compare with existing offline RL methods with the same metrics they have adopted. Following previous approaches (e.g. RvS[10], DD [1]), we train and evaluate our methods with multiple (e.g. 5) seeds, and calculate the mean and standard deviation across seeds for each dataset.
>
> >***In section 4.3, did the authors remove both the goal and the bottleneck inputs? Or just the bottleneck ones? Or just the goal?***
>
> Thanks for pointing it out! In section 4.3, we indeed just remove the goal from TrajNet during the bottleneck computation. We retain both the bottleneck and the goal as inputs to PolicyNet. The ablation thus studies the importance of goal-conditioning for the bottleneck representation to serve as an “implicit planner”. We will update section 4.3 to clarify.
>
>
> [a] Kostrikov et al. "Offline reinforcement learning with implicit q-learning." arXiv preprint arXiv:2110.06169 (2021).
>
> [b] Yang, Rui, et al. "Rethinking Goal-Conditioned Supervised Learning and Its Connection to Offline RL." International Conference on Learning Representations. 2021.
>
> [c] Hejna, Joey, Jensen Gao, and Dorsa Sadigh. "Distance Weighted Supervised Learning for Offline Interaction Data." arXiv preprint arXiv:2304.13774 (2023).
>
> [d] Ghosh, Dibya, et al. "Learning to Reach Goals via Iterated Supervised Learning." International Conference on Learning Representations. 2020.
>
> [e] Ma, Jason Yecheng, et al. "Offline goal-conditioned reinforcement learning via F-advantage regression." Advances in Neural Information Processing Systems 35 (2022): 310-323.

---

> > ### Comment · Reviewer_3wNA · 2023-08-16
> > **Reviewer Response**
> >
> > I would like to thank the authors for their response and detailed comments.
> >
> > > In our experiment, we regard IQL[a] as a robust value-based baseline ... we applied more rollouts for Kitchen and Locomotion, and found their performances remained stable when using 10 or more rollouts.
> >
> > I think it'd be better to include a thorough analysis and benchmarks on the choice of the number of rollouts for each scenario.
> >
> > > Yes, we indeed observed that performance may oscillate for the AntMaze environment. .... Our GCPC outperforms both baselines by large margins.
> >
> > It'd be great if you can include the Fig R1 to the main text.
> >
> > > Following previous approaches (e.g. RvS[10], DD [1]), we train and evaluate our methods with multiple (e.g. 5) seeds, and calculate the mean and standard deviation across seeds for each dataset.
> >
> > Let me disagree on this one. When the tide is going the wrong direction, we do not follow it but try to move in the right direction. Besides this should be just a matter of recalculating a few metrics (no need to rerun the experiments if you have the values). You can always include the less good but more widely used metric in the supplementary. Please do adapt this or provide an analysis (e.g., in the supplementary) on why in your specific cases mean/std-variance is better suited than median/percentiles. A response of the type "people usually do this 'bad' practice, so we continue it" it's not good enough.
> >
> > Other than that, I am pleased with the response by the authors.

---

> > > ### Author Response · Authors · 2023-08-16
> > > **Authors response**
> > >
> > > Dear reviewer 3wNA,
> > >
> > > Thank you so much for your feedback!
> > >
> > > > I think it'd be better to include a thorough analysis and benchmarks on the choice of the number of rollouts for each scenario.
> > >
> > > Yes, we already have most of the results on the impact of number of rollouts for each environment, we will include the analysis in the final version (likely in appendix, due to space limit).
> > >
> > > > It'd be great if you can include the Fig R1 to the main text.
> > >
> > > Yes. We will include the Figure R1 (after making it more compact) in the main text.
> > >
> > > > Please do adapt this or provide an analysis (e.g., in the supplementary) on why in your specific cases mean/std-variance is better suited than median/percentiles.
> > >
> > > We acknowledge and fully agree that an analysis comparing the two metrics would be very useful. Compiling the results under the revised metrics would take time beyond the discussion phase deadline, as we would need to run the key baseline methods for proper comparisons. We would appreciate if you could allow us to report the analysis in the final version. Depending on the outcomes of our experiments and analysis, we would either report the new metrics in the main text, or provide an analysis on our rationale to use the original metrics in the appendix.
> > >
> > > We hope our response addresses your followup questions, thank you!
> > >
> > > Best,
> > >
> > > The Authors

---

### Author Rebuttal · Authors · 2023-08-10

# General Response to All Reviewers

We thank all reviewers for their insightful suggestions and constructive feedback. We are glad that the reviewers found our work to be “interesting”, “novel” (EZxU, tnif, 3wNA), offers a “nice” and “fresh” perspective on existing RL methods (tnif, EzxU), and that our experiments to be “extensive”, “thorough” and “well presented” (eFoi, 3wNA, EZxU). Below we summarize our findings in the newly added experiments as requested by the reviewers, and respond to their common questions:

1. We’ve presented the results of ablation experiments over 5 seeds on Antmaze and Kitchen (Tables R1, R2, and R3). The 5-seed results are consistent with the observations reported in the main paper and supplementary materials. Specifically, we observe that action-free goal-conditioned predictive coding yields significant improvements on Antmaze Large. As in Tables A3 and A4 of the supplementary, the impacts of masking schemes and goal-conditioning on Kitchen are less distinct, and most result differences fall within error bounds. We attribute this to the small dataset size of Kitchen, which we suspect leads to the overfitting problem on TrajNet.

2. We’ve provided two additional goal-conditioned supervised learning baselines, WGCSL[a] and DWSL[b], on Antmaze and Kitchen (Table R4). We adopt the implementation and the recommended hyperparameters as shared by the authors of DWSL, and use the same evaluation setup as GCPC. We observe that GCPC outperforms both goal-conditioned baselines by large margins on both environments.

3. We’ve evaluated our method on CALVIN[c], another long-horizon manipulation environment, featuring four target subtasks and a substantial collection of task-agnostic trajectories. We use the same evaluation protocol and 15 trajectory rollouts. As shown in the table below, our method outperforms IQL[d], POR[e], and RvS-G by large margins.

|        | IQL[d]       | POR[e]        | RvS-G         | GCPC          |
|--------|--------------|---------------|---------------|---------------|
| CALVIN | 7.8 (± 17.6) | 12.4 (± 18.6) | 20.6 (± 13.4) | 48.9 (± 11.6) |

[a] Yang, Rui, et al. "Rethinking Goal-Conditioned Supervised Learning and Its Connection to Offline RL." International Conference on Learning Representations. 2021.

[b] Hejna, Joey, Jensen Gao, and Dorsa Sadigh. "Distance Weighted Supervised Learning for Offline Interaction Data." arXiv preprint arXiv:2304.13774 (2023).

[c] Mees, Oier, et al. "Calvin: A benchmark for language-conditioned policy learning for long-horizon robot manipulation tasks." IEEE Robotics and Automation Letters 7.3 (2022): 7327-7334.

[d] Kostrikov et al. "Offline reinforcement learning with implicit q-learning." arXiv preprint arXiv:2110.06169 (2021).

[e] Xu, Haoran, et al. "A policy-guided imitation approach for offline reinforcement learning." Advances in Neural Information Processing Systems 35 (2022): 4085-4098.

---

### Author Response · Authors · 2023-08-21
**Official Comment by the Authors**

Dear reviewers and area chairs,

As the reviewer-author discussion phase is coming to an end, we would like to say thank you, and that we truly appreciate your constructive feedback, which helps substantially improve our submission. Your detailed and timely feedback during the discussion phase means a ton for us.

We are glad that our rebuttal and the additional experiments requested by the reviewers have addressed most of the concerns from the reviewers, and we will incorporate the suggested changes (e.g. improving the presentation of experimental section and the limitations, moving the discussion on "implicit planning" completely into appendix, incorporating the suggested evaluation metrics, incorporating the new experimental results, etc.) in the final version. Thank you!


Best,

The Authors

---

### Decision · Program_Chairs · 2023-09-21

**Decision:**

Accept (poster)

**Comment:**

This paper investigates how supervised sequencing models can/should be used for decision making, including different representation learning strategies from different masking of sequenced, and an empirical analysis of how those representations help control performance on various simulated control tasks. This paper also proposes their own method Propose Goal conditioned predictive coding (GCPC).

This paper had mixed reviews, with valid concerns and points of praise. And after reading the paper and the reviews, I lean accept.

This paper's main strength is in the empirical analysis, the various comparison of various masking strategies and representations towards useful to control, and comparison to relevant recent baselines, which I believe is novel and of value to the NeurIPS community. The authors have diligently responded to the many valid reviewer concerns, which have mainly been about clarity and additional experiments. I highly recommend the authors to follow through on these promised changes and include them in their camera ready, including Reviewer tnif's concerns to downplay claims of method superiority when compared performance is within error bounds. Without all these changes I would have leaned reject, so since the changes are critical towards my flip to an "accept" rating, and I will check for these changes in the camera ready copy, so please include them!

PS: line 73 typo: “reinformenct”